# Benthos as a key driver of morphological change in coastal regions

Peter Arlinghaus[1], Corinna Schrum[1,2], Ingrid Kröncke[3,4], Wenyan Zhang[1]

[1]Institute of Coastal Systems - Analysis and Modeling, Helmholtz-Zentrum Hereon, Geesthacht, Germany,

[2]Center for Earth System Sustainability, Institute of Oceanography, Universität Hamburg, Hamburg, Germany

[3]Institute for Chemistry and Biology of the Marine Environment (ICBM), Carl von Ossietzky University, Oldenburg, Germany

[4]Department for Marine Research, Senckenberg am Meer, Wilhelmshaven, Germany

*Correspondence to*: Peter Arlinghaus (peter.arlinghaus@hereon.de), Wenyan Zhang (wenyan.zhang@hereon.de)

**Abstract.** Benthos has long been recognized as an important factor influencing local sediment stability, deposition and erosion rates. However, its role in long-term (annual-to-decadal scale) and large-scale coastal morphological change remains largely speculative. This study aims to derive a quantitative understanding of the importance of benthos in the morphological development of a tidal embayment (Jade Bay), as representative for tidal coastal regions. To achieve this, we firstly applied a machine learning-aided species abundance model to derive a complete map of benthos (functional groups, abundance and biomass) in the study area, based on abundance and biomass measurements. The derived data were used to parameterize the benthos effect on sediment stability, erosion/deposition rates, and hydrodynamics in a 3-dimensional hydro-eco-morphodynamic model, which was then applied to the Jade Bay to hindcast morphological and sediment change for 2000-2009. Simulation results indicate significantly improved performance with benthos effect included. Simulations including benthos show consistency with measurements regarding morphological and sediment changes whilst abiotic drivers (tides, storm surges) alone result in a reversed pattern in terms of erosion and deposition contrary to measurement. Based on comparison among scenarios with various combinations of abiotic and biotic factors, we further investigated the level of complexity of hydro-eco-morphodynamic models that is needed to capture long-term and large-scale coastal morphological development. The accuracy in parametrization data was crucial for increasing model complexity. When the parametrization uncertainties were high, increased model complexity decreased model performance.

## 1. Introduction

Benthos includes flora such as sea grass, kelp and salt marsh species, which predominately stabilizes sediment (Corenblit et al., 2011; Zhang et al., 2012; Zhang et al., 2015) and fauna with more complex behaviors that can stabilize or destabilize sediment (Backer at al., 2010). Benthic in- and epifauna actively reworks sediment in order to increase the availability of resources for themselves (Jone et al., 1994; Meadows et al., 2012), and plays a critical role in modifying sediment properties such as grain size, porosity, permeability and stability at local scales in coastal environments (Backer et al., 2010 Arlinghaus et al., 2021; Murray et al., 2008).

The different behaviors of benthos and consequent impacts on sediment have been described in numerous studies and literature reviews (Arlinghaus et al, 2021; Andersen and Pejrup, 2011; le Hir et al., 2007). Major benthos behaviors include biomixing (Lidqvist et al., 2016; Queiros et al., 2013, Meyer et al., 2018, Weinert et al., 2022), bioirrigation (Wrede et al., 2017), biodeposition and -resuspension (Cozzoli et al., 2019; Graf and Roseberg, 1996), faecal pellet production (Andersen and Perjup 2011; Grant and Daborn, 1994; Troch et al., 2008) and biofilm stabilization (Le Hir et al., 2007; Stal et al., 2010). All these ways in which benthos changes and modifies the sediment directly or indirectly is termed bioturbation (Meysmann et al., 2007). The impacts of bioturbation on sediments can individually or accumulatively lead to dramatic local morphological changes as demonstrated by

defaunation experiments (Volkenborn and Reise, 2006; Volkenborn et al., 2008; Montserrat et al., 2008). However, most studies are limited to small temporal and spatial scales and it remains unclear whether such small-scale benthos-sediment interactions could affect long-term (annual-to-decadal scale) and large-scale (km-to-basin scale) coastal morphological change.

Over the past three decades, increasing efforts have been dedicated to upscale the impacts of benthos-sediment interactions to larger scales through the use of numerical modeling (Arlinghaus et al., 2021). Results indicate that benthos can induce erosion that is in the same order of magnitude as hydrodynamics (Wood and Widdows, 2002; Lumborg et al., 2006; Arlinghaus et al., 2022) and causes redistribution of sediments at large spatial scales, e.g. across tidal basins (Borsje et al., 2008) and coastal bays (Nasermoaddeli et al., 2017). Fine-grained, muddy sediments are especially sensitive to benthos impacts (Paarlberg et al., 2005; Knaapen et al., 2003; Smith et al., 1993). However, almost all modeling studies applied at large-scales are limited to qualitative results (Arlinghaus et al., 2021). Following the concept of Desjardins et al. (2018), numerical models can be categorized into three types corresponding to successive development stages, namely explorative, explanatory and predictive models. In explorative hydro-eco-morphodynamic models, processes and their parameterizations are varied within a certain range, creating an ensemble of possible final states to estimate and explore the impact range of a driver, e.g. benthos, on morphological evolution. In explanatory models, a certain final state is known and the model parameters are tuned in order to hindcast the change of the system from an initial state to the final state as accurate as possible, so that the simulation results can be used to understand the magnitude and relative importance of the involved processes contributing to the final state. Most hydro-eco-morphodynamic models are still at the explorative stage and have yet to reach the explanatory stage, and the reason is manifold. In general, benthic physical and biological processes are highly complex, involving many feedback loops and boundary conditions with large variability (Oreskes et al., 1994; French et al., 2015; Larsen et al., 2016), e.g. many biophysical functions such as the formation of biofilm and its impact on sediment stability remain still poorly understood (Stal, 2010; Van Colen et al., 2010; Chen et al., 2017). Interactions between different functional groups of benthos and between benthos and seabed morphology are important in coastal morphodynamics (Murray et al., 2008; Marani et al., 2010; Corenblit et al., 2011; Reinhardt et al., 2010; Zarnetske et al., 2017) but have rarely been incorporated in large-scale modeling (Arlinghaus et al., 2022, Brückner et al., 2021). Shortage of continuous field monitoring data (e.g. mapping of benthos and seabed morphology) with long-term coverage impedes a process-based understanding and mathematical description of benthic biophysical functions (Arlinghaus et al., 2021).

Explanatory models represent an intermediate stage of model development from exploratory toward predictive modeling (Desjardins et al., 2018). This study presents an effort to this end in hydro-eco-morphodyamic modeling. For this purpose, the Jade Bay, a tidal embayment located in the German Wadden Sea, was chosen to test the model. The reason for choosing the Jade Bay is that extensive datasets for both morphological evolution and biological parameters are available for the area, providing a unique opportunity for an explanatory modeling investigation.

Tidal embayments such as the Jade Bay are commonly found worldwide (Haas et al., 2017). They are among the most productive ecosystems in the Earth surface providing a variety of ecosystem functions (Mitsch and Gosselink 2007) and serve as important habitats for marine lifeforms (Levin et al., 2001). On the other hand, they are

commonly utilized for fishing, navigation and tourism and endure strong population pressure (Duong et al., 2016).

Depending on the effects of different biotic and/or abiotic drivers, tidal embayments may persist for centuries, be filled up or closed (Haas et al., 2017), or be drowned (Plater and Kirby, 2011). Thus, understanding the morphodynamics of these systems is crucial for coastal mitigation and adaptation in response to climate change and human use.

In this study, an elaborate hydro-eco-morphodynamic model is used to hindcast the morphological development of the Jade Bay from 2001 to 2009. Jade Bay benthos data include infauna (>0.5 mm) and seagrass. By incorporating the impacts of these two types of benthos, we aim to address the following specific questions:

1. To what extent do benthos accounts for the observed changes in the morphology and sediment
composition in the study area? and
2. What are the individual and combined impacts of different functional groups on morphological development?

## 2. Study Area

Jade Bay is located in the inner part of the German Wadden Sea and connected to the outer part through a deep
(>15 m) tidal inlet (Fig. 1). The tidal inlet and the Jade Bay have a combined length of approx. 36 km and vary in width between 4 and 15 km, covering around 370 km², with 160 km² inside the bay, and about 60 % of which is comprised of tidal flats (Lang et al., 2003). The Jade Bay is a meso-tidal system with a tidal range of ca 3.7 m (Svenson et al., 2009). The water depth of the main channel reaches up to 20 m below the mean sea level. The main channel penetrates Jade Bay and branches into three major basin channels which are permanently inundated
(Stenckentief, Vareler Fahrwasser, Ahne, see Fig. 1a). The intertidal area has a mean water depth of 2.07 m during high tide (Von Seggern, 1980). Tidal currents transport an average volume of 0.4 km³ per tidal cycle with speed exceeding 1.5 m/s in the channels (Götschberg and Kahlfeld, 2008). A training wall guides tidal currents, leading to finer sediments towards the western and southern parts of the bay (Linke, 1939, Götschberg and Kahlfeld, 2008). The central part of the channel is characterized by medium to coarse sands, while towards the banks fine sands
with increasing mud content are found (Reineck and Singh, 1967). Three bed types can be distinguished: sandflats, mudflats and mixed. The bay is inhabited by abundant benthic fauna and seagrass meadows (*Zostera noltii*). In terms of biomass the most abundant organisms are Bivalvia (*Cerastorderma edule, Macoma balthic*), Gastropoda (*Peringia ulvae*) and Polycheats (*Arenicola marina, Hediste diversicolor, Tubificoides benedii*) with a spatially averaged biomass of 20 g C m$^{-2}$ according to Schückel et al. (2015a). Typical values of benthic biomass range
between 1-100 g C m$^{-2}$ in the Wadden sea (Beukema, 1974; Reise et al., 1994; Beukema and Dekker, 2020).

## 3. Methods
### 3.1 Machine learning-aided mapping of macrobenthos

According to the impacts of benthos on sediment dynamics and to achieve an appropriate level of model
complexity, benthos are sorted into functional groups. A functional group comprises species from different taxa that impact their environment in similar ways (Kristensen et al., 2012). In this study, benthos is categorized into four major functional groups, namely biomixers, stabilizers, accumulators, and seagrass. Biomixers and accumulators consist of macrobenthos while stabilizers are represented by a biofilm which is mainly assembled

by microphytobenthos (MPB) of all contributing species. The seagrass present in Jade Bay belongs to the species

*Zostera noltii* (Adolph, 2010).

The existing field data set provides macrobenthos abundance in the inter-tidal area and abundance plus biomass for the subtidal area at 160 stations in the Jade Bay (Senckenberg, Schückel and Kröncke, 2013; Schückel et al., 2015b). Based on the intertidal abundance values and biomass averages from the subtidal, the intertidal biomass could be calculated (Fig. 2b-f). The total measured biomass in the Jade Bay is dominated by a few species which

are widely distributed in the area. Since the metabolic rate of biomixers is a useful indicator for bioturbation intensity (Cozzoli et al., 2019), which scales with biomass, we focus on five dominant species which make up 95% of benthos biomass in the area, namely the mussels *Cerastoderma edule* (accumulator) and *Macoma balthica* (accumulator and biomixer), the snail *Peringia ulvae* (biomixer) and the worms *Hediste diversicolor* (biomixer) and *Tubificoides benedii* (biomixer). Complete mapping of benthos for the entire Jade Bay is done by extrapolation

from 160 field stations. Species distribution modeling (SDM) is commonly used for this purpose which produces probabilities of species occurrence. Various methods have been applied, spanning from statistical methods to machine learning (Waldock et al., 2022). Species abundance modeling (SAM) is developed from SDM and has an increased solution space, since the output represents decimal values covering the whole range of measured abundance spectrum or biomass spectrum respectively. Existing studies show best results using decision trees

(Luan et al., 2020; Waldock et al., 2022). For this reason we adopted a decision tree-based SAM to generate a complete map of benthos in the study area. Detailed description of the method and analysis of the applied dataset are provided in the supplementary material.

Six predictor variables at the stations, namely temperature, salinity, Chl-a content, inundation time, shear stress and mud content were used. The first three were derived via image analysis of the plots from the Jade Bay SDM

results by Singer at al. (2016) and the latter three were extracted from the hydrodynamic model results. Abundance and biomass of the five dominant species are target variables. For each of the species a separate regression tree model was run for the Jade Bay area. In addition, the SAM model was extended to cover the inner and outer Jade. However, in this area there is no benthos field data for model validation and the number of predictor variables is reduced to three (mud content, shear stress and inundation time). Based on the field data, two SAM models were

applied for each species, one for abundance and one for biomass, in order to calculate the mean individual biomass which is needed for the parametrization of benthos impacts on sediment. We used 90% of the species data points for model training and the rest 10% to test the model performance.

Although the field dataset of benthos abundance and biomass is uniquely comprehensive for a tidal basin in the Wadden Sea, seasonal variations were not covered. To take into account seasonal variations of benthos impact, a

simple sinusoidal function describing the change of biomass and related bioturbation intensity (see details in section 3.2.1) was used in some of the model experiments described in Table 3.

## 3.2 Mathematical description of benthos impact

Impacts of benthos on sediment are formulated through scaling functions between benthos abundance/biomass and model parameters for sediment dynamics, namely the critical shear stress for erosion $\tau_c$ $(Pa)$, the erosion rate

$E_r$ $(\frac{kg}{m^2 s})$, the sediment settling velocity $W_{sed}$ $(\frac{mm}{s})$ and hydrodynamic parameters for turbulence and bottom shear stress. For sediment erosion, the general approaches by Knaapen et al. (2003) for $\tau_c$ and Paarlberg et al. (2005) for $\tau_c$ and $E_r$ are applied. An abiotic critical shear stress for erosion $\tau_c^0$ and erosion rate $E_r^0$ are scaled by dimensionless biomixing functions $p_d$, $g_d$ and stabilization functions $p_s$, $g_s$, respectively, which depend on

abundance $A$ (number of individuals) and biomass $B$ ($mg$ ash free dry weight (AFDW)) of these two functional groups:

$$\tau_c = \tau_c^0 \cdot p_d(B,A) \cdot p_s(B,A) \tag{1}$$

$$E_r = E_r^0 \cdot g_d(B,A) \cdot g_s(B,A) \tag{2}$$

Changes in hydrodynamics by the effect of seagrass are incorporated using the *submerged aquatic vegetation model* (SAV) of SCHISM (Zhang et al., 2016) and changes in $W_{sed}$ by the effect of accumulators are applied according to a filter feeder ingestion rate model (US Army Corps of Engineers, 2000). Both are explained in following sections. No direct control between different functional groups is considered in the presented simulations.

### 3.2.1 Biomixers

The main effect of biomixers is sediment destabilization. However, biomixing macrobenthos can also increase sediment stability in certain conditions of metabolic rate, bottom shear stress and sediment composition (Cozzoli et al., 2019), which is attributed to hardening of mucus excreted during locomotion (Orvain, 2002; Le Hir et al., 2007). In our model, the formulae from Cozzoli et al. (2019) are adopted to relate biomixing effect with the overall metabolic rate $M_{TOT}$ ($mW$). In this study measurements of the total eroded sediment per unit area in a given time, $R_{TOT}$ ($\frac{g}{m^2}$), were taken. Assuming that the erosion rate ($\frac{kg}{m^2 s}$) over the given time is constant it can be described by:

$$R_{TOT} = \frac{a}{1+\exp\left(\frac{b-\tau_b}{c}\right)}, \tag{3}$$

where the factors $a$ ($\frac{g}{m^2}$) and $b$ ($Pa$) are related to $M_{TOT}$ and $B$, $c$ ($Pa$) is an empirical constant, and $\tau_b$ is the bottom shear stress. In order to calculate $M_{TOT}$, measurements from Cozzoli et al. (2019) (Table 1) are used to estimate the individual metabolic rate ($M_{Indv}$ ($mW$)) from the individual biomass ($B_{Indv}$ ($mg$ AFDW)):

$$M_{Indv} = 0.0067 \cdot B_{Indv}^{0.835} \tag{4}$$

The SAM results for abundance and biomass are then used to calculate the mean individual biomass, which is fed into Eq. (4) to derive $M_{Indv}$ and total metabolic rate $M_{TOT}$ by multiplying with the abundance $A$. The derived value of $M_{TOT}$ is then used to calculate the factors $a$ and $b$ under biomixing impact ($a_{bio}$ and $b_{bio}$):

$$a_{bio} = 41.67 \cdot (1 + M_{TOT})^{0.34} \cdot (1 + B_{Indv})^{-0.09}, \tag{5}$$

$$b_{bio} = 0.1 + 0.01 \cdot \log(1 + M_{TOT}). \tag{6}$$

The total eroded sediment under biomixing impact, $R_{TOT}^{bio}$, is calculated by feeding $a_{bio}$ and $b_{bio}$ into Eq. (3). The total eroded sediment under abiotic conditions $R_{TOT}^0$ is calculated based on the formulation given in Cozzoli et al. (2019) and is used to derive the biomixing function $g_d$:

$$g_d = \frac{R_{TOT}^{bio}}{R_{TOT}^0} \tag{7}$$

The other biomixing function $p_d$ is calculated following Brückner et al. (2021), which is also based on the data from Cozzoli et al. (2019). Abiotic ($\tau_c^0$) and biotic critical shear stress for erosion ($\tau_c^{bio}$) are defined based on the respective $\tau_b$ value at which a minimal erosion rate of 25 g m$^{-2}$ is reached. This is done by converting formula (3) into:

$$\tau_c = b - c \cdot log\left(\frac{a-R_{25}}{R_{25}}\right) \tag{8}$$

$\tau_c^0$ is calculated using $a_0, b_0$, and $c_0$ which are constants for the defaunated control experiments given in Table 1 in Cozzoli et al. (2019). For $\tau_c^{bio}$ $a_{bio}$, $b_{bio}$, and $c_0$ are used. $p_d$ is then calculated via:

$$p_d = \frac{\tau_c^{bio}}{\tau_c^0} \tag{9}$$

$g_d$ and $p_d$ are calculated by adding up all biomixing species considered in the SAM. For the Jade Bay, the derived values of $g_d$ and $p_d$ show a strong destabilizing effect on a vast part of the bay especially on the tidal flats, while
the subtidal area is mainly stabilized (Fig. S3, supplementary material).

Macrobenthic oxygen consumption rate may decrease by a factor of 10 during winter compared to summer (Glud et al., 2003; Renaud et al., 2007) and thus biomixing intensity may also decrease accordingly. To account for this seasonal variability, a multiplication factor for $M_{TOT}$ was introduced according to a sine function with a period of 1 year, reaching the maximum value of 1.0 in summer and the minimum of 0.1 during winter.

**3.2.2 Stabilizers**

The stabilization functions $p_s$ and $g_s$ are related to biofilm which is primarily built by microphytobenthos (MPB). According to measurements by le Hir et al. (2007) and Waeles et al. (2007), an increase of the critical shear stress for erosion ($\tau_c$) by a factor of 4 ($p_s = 4$) is implemented for the summer months (from June to September) when MPB is present. For the rest of the year a factor of one is used because MPB is mostly not present in winter and
has thus no effect ($p_s = 1$). Erosion rate ($E_r$) is assumed to be unaffected by MPB, thus $g_s$ is set to 1 as a constant.

**3.2.3 Accumulators**

The presence of accumulators (mainly suspension and filter feeders) such as mussels effectively increases the settling velocity of sediment particles in the bottom water layer. The magnitude of resulting bio-deposition rate of sediments depends on the filtration rate and ingestion rate $I\left(\frac{L}{mg}\right)$ of accumulators which scales with biomass
$B_{acc}\left(\frac{mg\ AFDW}{m^2}\right)$. In this study, a simplified version of the filter feeder model from the US Army Corps of Engineers (2000) excluding the temperature effect was applied. Sediment particle settling velocity in the bottom most water layer ($W_{sed}$) is modified by:

$$W_{sed} = W_{sed}^0 + I \cdot B_{acc} \tag{10}$$

where $W_{sed}^0$ represents the settling velocity without the effect of accumulators. Further details of the parametrization are provided in the supplementary material.

**3.2.4 Seagrass**

The impact of seagrass is incorporated by an additional drag term in the Reynolds-averaged Navier-Stokes
equation and an additional source term for turbulent kinetic energy and mixing length, following the implementation of Cai (2018). The magnitude of these terms depends on the canopy height $h\ (mm)$, the stem diameter $d\ (mm)$, stem density $N\left(\frac{1}{m^2}\right)$ and the drag coefficient for vegetation $c_D$. The parameters were chosen according to the vegetation cover and the common densities of $Z.\ noltii$ in the German Wadden Sea (Adolph, 2010) and are listed in the model setup section. Seasonal change of seagrass is not included in this study due to lack of
field data support for parameterization.

**3.3 Hydro-eco-morphodynamic numerical model**

The formula for benthos effect on sediment dynamics described in section 3.2 are integrated into a 3-dimensional modeling system SCHISM (Zhang et al., 2016) to simulate hydro-eco-mophodynamics. SCHISM solves the Reynolds-averaged Navier-Stokes equation on an unstructured horizontal grid employing a semi-implicit Galerkin

finite element method (FEM). Vertical velocities and transport is computed with a finite volume method (FVM) approach for a flexible number of vertical layer, allowing for transition between regions of different depth and resolution (Zhang et al., 2008). Turbulence closure is implemented according to the k-kl closure scheme described in Umlauf and Burchard (2003). The original SCHISM framework includes a sediment module (SED3D, Pinto et al., 2012) which does not take into account the impacts of benthos. Sediment is divided into multiple classes, each with characteristic parameters including grain size, density, settling velocity, erosion rate and critical shear stress for erosion. Cohesive and non-cohesive sediments are distinguished. Non-cohesive sediments (sands) can be transported in both suspension and bed-load depending on the shear stress and settling velocity, while cohesive sediment (clay, silt and organic detritus) is transported in suspension. Transport of each pre-defined sediment class is computed independently.

### 3.3.1 Model setup for the study area

The model domain spans roughly from 53°23'N 8°35'E to 53°53'N 7°46'E (Fig. 1a). It is covered by unstructured triangular elements with a spatial resolution of approx. 800 m in the outer Jade and an increasing resolution toward the Jade Bay, with a resolution of approx. 200 m inside the bay. The vertical plane is divided into 11 sigma layers. The open boundary is forced by 15 tidal constituents (M2, K1, S2, O1, N2, P1, SA, K2, Q1, NU2, J1, L2, T2, MU2, 2N2) extracted from the global ocean tide atlas FES2014 (Florent et al., 2021) as well as observed storm surges which were implemented in terms of water level changes (supplementary material). These changes are based on measurements at a gauge station (Lighthouse Alte Weser) located at the open boundary (Fig. 1a). Discharge is specified for the Weser River at the south east boundary of the modeling domain according to Galbiati et al. (2008). Two sediment classes which are dominant in the study area (Fig. 1b) are included, namely fine sands with an initial settling velocity ($W_{sed}^0$) of 1 mm s$^{-1}$ and mud with an initial settling velocity ($W_{sed}^0$) of 0.02 mm s-1. A constant mud concentration of 40 mg l$^{-1}$ is specified at the open boundary according to Pleskachevsky et al. (2005). Seasonal variability in suspended sediment concentration (SSC) at the open boundary were not implement due to the lack of measurement data. Turbidity and sediment concentration measurements from the Jade Bay typically cover one or a few points measured over one or a few tidal cycles (Götschenberg and Kahlfeld, 2008; Becker, 2011) while longer and larger scale measurements were absent. SSC in the presented simulations are in the same range as measurements from the Jade Bay (Becker, 2011) and comparable to another simulation study in Jade Bay (Kahlfeld and Schüttrumpf, 2006). A map of simulated SSC is provided in the supplementary (Fig. S7).

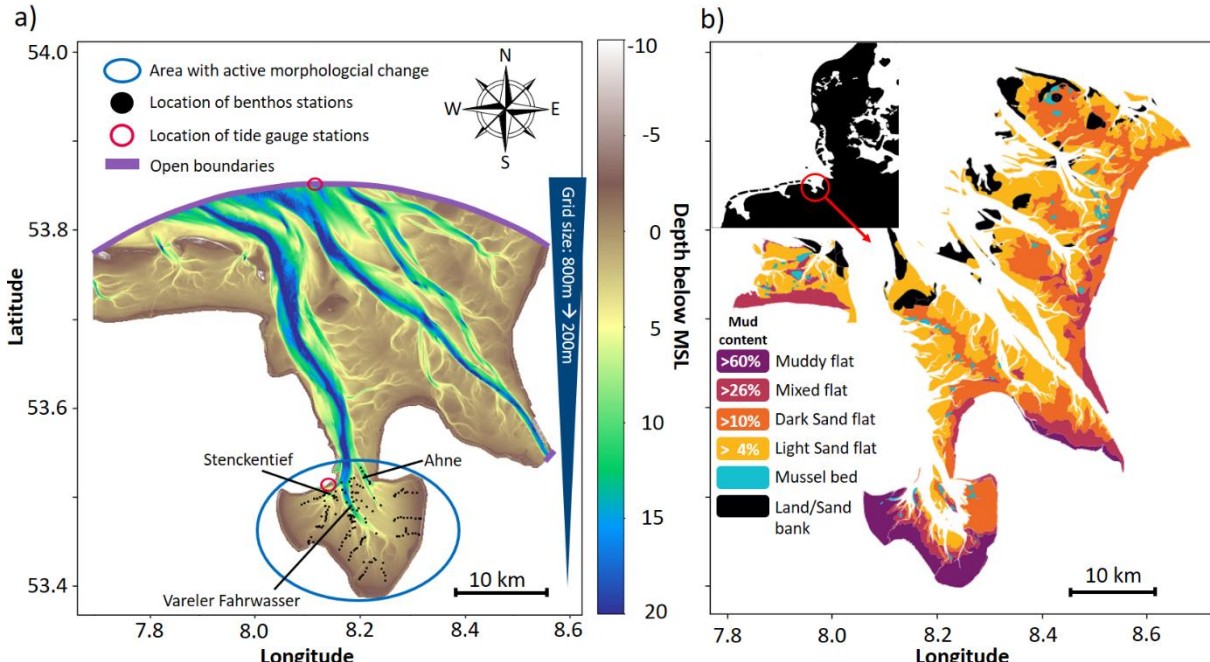

**Figure 1. (a) Computational domain and its open boundary, including the initial morphology at 2001, the location of benthos data and tide gauge stations; (b) Distribution of sediment types including land and mussel beds (Meyer and Ragutski, 1999).**

Datasets from various sources are used to initialize, parametrize and validate the model. A brief summary of these datasets is given in Table 1. The model is used to hindcast the change of morphology and sediment composition in the Jade Bay from July 2001 until December 2009. The measured morphology in 2001 serves as the initial condition. There are no sediment property measurement for periods around 2001, therefore measured data from 1996 (Fig. 1b) were used to specify the initial mud and sand contents. Default model parameters representing abiotic conditions are listed in Table 2.

**Table 1.** Data sources used for model initialization (Init.), parameterization (Param.), and model validation (Valid.).

| Type | Use | Time | Description | Source/Provider |
|------|-----|------|-------------|-----------------|
| **Benthos** | Init. | 2009 | Abundance and biomass at 160 field stations | Senckenberg, Kröncke and Schückel (2013), Schückel et al. (2015) |
| **Benthos** | Param. | - | Laboratory erosion measurements with different species at different densities | Cozzoli et al., 2019 |
| **Benthos** | Param. | - | Filter feeding rate for accumualtors | US Army Corps of Engineers, (2000) |
| **Benthos** | Param. | - | Estimated MPB impact | Le Hir et al., 2007 |
| **Benthos** | Param. | - | Seagrass impact on hydrodynamics | SAV module of SCHISM, Adolph (2010) |
| **Sediment** | Init. | 1996 | Sediment map | Meyer and Ragutski (1999) |

| Sediment | Valid | 1996-2009 | Map of sediment change | Ritzmann and Baumberg (2013) |
|---|---|---|---|---|
| **Forcing: tides** | Init. | 2001-2009 | Finite element global ocean tide atlas | FES2014 Florent et al., 2021 |
| **Forcing: storms** | Init. | 2001-2009 | Observed water elevation data at the gauge station Lighthouse Alte Weser | Wasserstraßen- und Schifffahrtsverwaltung des Bundes (WSV) |
| **Water level** | Valid. | 2001-2009 | Observation data at the gauge station Wilhelmshaven | Wasserstraßen- und Schifffahrtsverwaltung des Bundes (WSV) |
| **Morphology** | Init. + Valid. | 2001-2009 | High-resolution morphology of the German Bight | Sievers et al., (2020) |

**Table 2.** Configuration of default model parameters for abiotic conditions.

| Parameter | Configuration |
|---|---|
| $h$ | 25 cm |
| $d$ | 0.2 cm |
| $N$ | 400 m$^{-2}$ |
| $c_D$ | 1.13 |
| $\tau_c^0$ | 0.2 Pa |
| $E_r^0$ | $2 \cdot 10^{-5}$ s m$^{-1}$ |
| $E_r^{10}$ | $2 \cdot 10^{-4}$ s m$^{-1}$ |
| $W_{sed,mud}^0$ | $2 \cdot 10^{-5}$ m s$^{-1}$ |
| $W_{sed,sand}^0$ | $1 \cdot 10^{-3}$ m s$^{-1}$ |

In order to disentangle the impacts of benthos, including effect of individual functional groups and combined effect of all functional groups, and abiotic drivers on morphological and sediment change of the study area, a total of 27 different model experiments have been performed (Table 3). The experiments were designed to include different levels of complexity in the variability of physical forcing (e.g. with and without storms) and benthos (e.g. with and without seasonality). In addition, an increased erosion rate was applied to some experiments excluding biomixers

for comparability reasons. Biomixers strongly enhance SSC which leads to an increase of the impact of other functional groups such as accumulators. To achieve comparable SSC levels in simulations excluding biomixers, the basic erosion rate (E0) was increased by a factor of 10 (E10) which helps to distinguish the effects of certain functional groups from scenarios with all benthic groups included.

**Table 3**. Model experiments are designed for a combination of different physical forcing and functional groups which are abbreviated by *mix* (biomixers), *acc* (accumulators), *sta* (stabilizers), *gra* (seagrass), *all* (inclusion of all functional groups) and *abio* (abiotic model run without consideration of any benthos effect). Seasonal variations of benthos impact are abbreviated by *no*/<abbreviation of specific functional group> if they were excluded/included. Hydrodynamic forcing excluding/including storm surges are abbreviated by T / TS and a

default erosion rate / an erosion rate scaled by a factor of 10 are abbreviated by 1 / 10. The experiments are named by combination of the different model features separated by an underscore and read as: **Modeled functional groups_Seasonality_Hydrodynamics_Erosion Rate**. For example in the model experiment acc_acc_TS_10 accumulators are the simulated functional group, seasonality of accumulators was considered, both tides and storm surges were considered as hydrodynamic forcing and the erosion rate was scaled by a factor of 10.

| | E0 | E0 + Storm | E0 + Storm + Seasonality | E0 + Storm + Seasonality all | E10 | E10 + Storm | E10 + Storm + Seasonality |
|---|---|---|---|---|---|---|---|
| All benthos | all_no_T_1 | all_no_TS_1 | all_mix_TS_1 | all_all_TS_1 | -- | -- | -- |
| Biomixers | mix_no_T_1 | mix_no_TS_1 | mix_mix_TS_1 | -- | -- | -- | -- |
| Stabilizers | sta_no_T_1 | sta_no_TS_1 | sta_sta_TS_1 | -- | sta_no_T_10 | sta_no_TS_10 | sta_sta_TS_10 |
| Accumulators | acc_no_T_1 | acc_no_TS_1 | acc_acc_TS_1 | -- | acc_no_T_10 | acc_no_TS_10 | acc_acc_TS_10 |
| Seagrass | gra_no_T_1 | gra_no_TS_1 | -- | -- | gra_no_T_10 | gra_no_TS_10 | -- |
| Abiotic drivers only | abio_no_T_1 | abio_no_TS_1 | -- | -- | abio_no_T_10 | abio_no_TS_10 | -- |

## 4. Results

### 4.1 Mapping of benthos

To assess the performance of the decision tree-based SAM model, the measured data were split into training and validation datasets. The training dataset was used for training the model and the validation dataset was checked against the resulting estimations of biomass and abundance. The performance of the SAM varies among the selected species. For the majority of the points, the estimated value deviates from the measured value by less than 20% (Fig. S2, supplementary). Biomass and abundance distributions of all five species are shown in Fig. 2b-f. For stabilizers, biofilm built by MPB is considered, which is only distinguished by presence or absence in the field data. We applied a formulation relating the growth of MPB-based biofilm to the inundation period and mud content following the studies by Widdows and Brinsley (2002) and Daggers et al. (2020). In the Jade Bay, only the western and southern parts are inhabited by extensive biofilms (Fig. 2a).

Seagrass distribution in the Jade Bay is described for the years 2000-2008 in Adolph (2010) with vegetation density between 5-40% for the dominant species *Zostera noltii* (Fig. 2a).

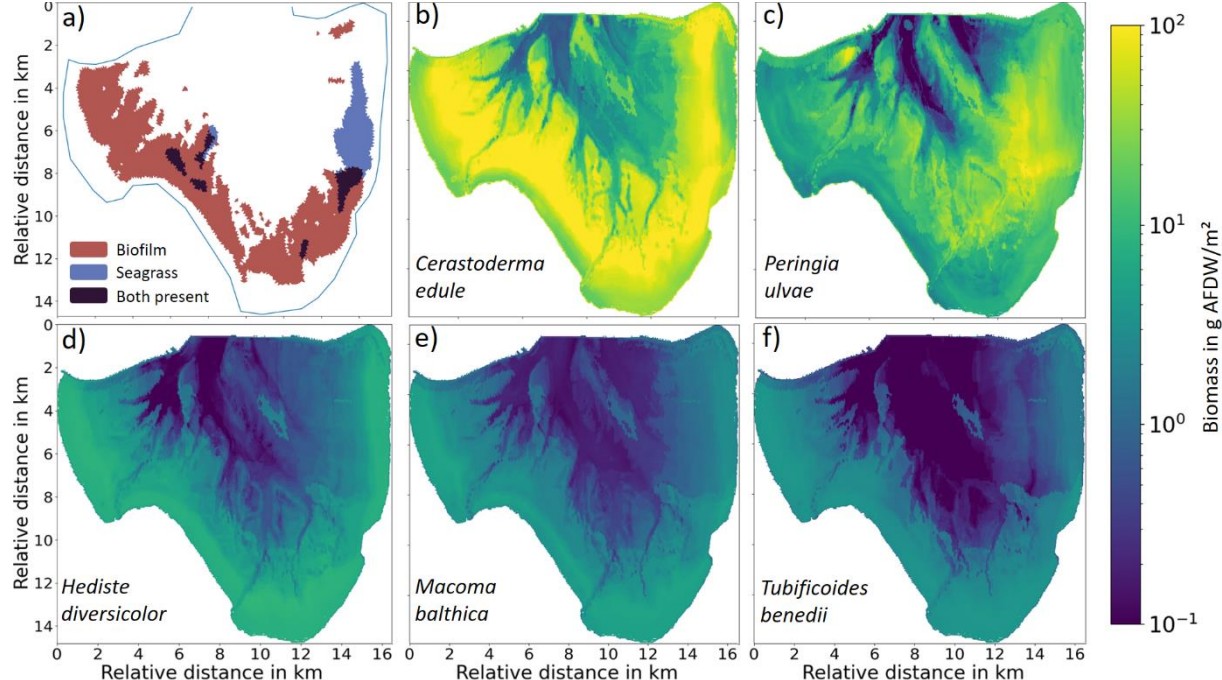

**Figure 2. (a) Presence of stabilizers and seagrass according to Adolph (2010); (b)-(f) Modeled biomass distribution of the five dominant benthic faunal species.**

**4.2 Assessment of hydro-eco-morphodynamic model performance**

Simulated time series of water level in all experiments are quite similar, and exhibit differences only during storm periods between the experiments with and without storms. Comparison with measured water level at a tide gauge station in Wilhelmshaven, which is located at the inlet of the Jade Bay, shows a satisfactory model performance

(Fig. 3). Taking the reference experiment abio_no_TS_10 as example, the standard deviation is 1.34 m for the data measured at the gauge station compared to 1.33 m derived from model results. For the tide gauge station at the Lighthouse Alte Weser the values are 1.03 m and 0.99 m respectively. The correlation coefficient between modeled water elevation and measured data is 0.98 at Wilhelmshaven and 0.96 at Alte Weser station (Fig. 3b).

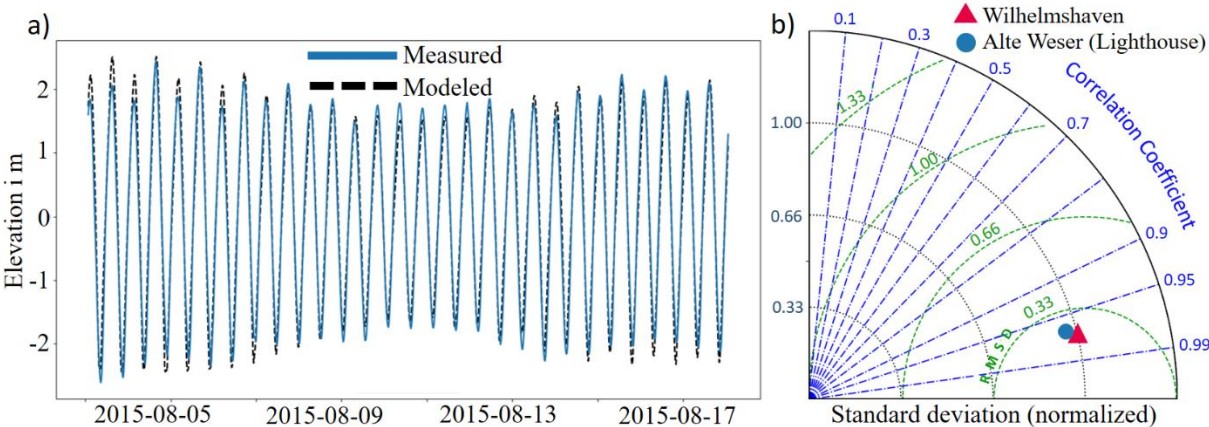

**Figure 3. (a) Modeled and measured water elevation at the tide gauge station in Wilhelmshaven. (b) Comparison between model results and measurement at the gauge stations in Wilhelmshaven and the Lighthouse Alte Weser in a Taylor diagram.**

The simulated change of sediment composition and morphology in all experiments are compared and evaluated.

Firstly, simulation results are evaluated against observed changes to rank the performance of the experiments. Then, the impact of individual functional groups and their combined effect is analyzed based on the model results.

In addition, the level of complexity of hydro-eco-morphodynamic models that is needed to capture long-term and large-scale coastal morphological development is investigated.

In order to minimize the effect of uncertainty in measurements, only the grid cells where the measured morphological change exceeds the standard deviation of difference between the 2001 and 2009 field data were chosen for the comparison in Fig. 4. Two indicators, namely the RMSE and the cosine similarity between the modeled and measured morphological change, were calculated for each of the experiments and shown in Fig. 4. The RMSE (Fig. 4a) shows the best model performance in the group of experiments (all_x) which take into account

the combined effect of all benthos functional groups, followed by the group of experiments (mix_x) which include the effect of biomixers only. The experiments (acc_x) which include only the accumulators show a better performance than the reference experiments (abio_x) which consider only abiotic drivers, whist the experiments which include only seagrass (gra_x) or stabilizers (sta_x) do not show noticeable improvement compared to abiotic scenarios. The difference in the RMSE between the model results with the best and the worst performance is about

15 cm, being about 150% of the average and 35% of the standard deviation of morphological change for the entire Jade Bay from 2001 to 2009. It is worth noting that within the group of experiments (all_x) which include all functional groups, better model performance is gained when storms are included (all_no_TS_1) and seasonality of the dominant functional group, namely the biomixers, is included (all_mix_TS_1). However, model performance decreases when seasonality of all functional groups is considered (all_all_TS_1). The decrease of

model performance due to inclusion of seasonality is also seen in other experiments which consider only one functional group, whilst an inclusion of storms only slightly enhances or does not affect the performance of these experiments. On the other hand, an increase of erosion rate by a factor of 10 improves the performance of the simulations which considers only abiotic drivers (abio_x) and those which include only one functional group (gra_x, acc_x, sta_x), although their performance is still worse than the experiments with combined effect of all

functional groups (all_x).

The cosine similarity between the modeled and measured morphological change provides further evaluation of the model performance in capturing the change in the main topographic units. It is a measure of similarity between two non-zero vectors which can be derived from the Euclidean dot product. In our evaluation, the cosine similarity is calculated for the main tidal channels (Stenckentief, Vareler Fahrwasser, Ahne, see Fig. 1). Results (Fig. 4b)

show that in the experiments with all benthos (all_x) and with inclusion of only biomix (mix_x), a positive correlation is found, suggesting that the modeled change is consistent with the measured change. On the contrary, a negative correlation is found in all other experiments, suggesting that an opposite pattern is produced in the model results compared to measurement. It is worth noting that an increase of erosion rate by a factor of 10 further strengthens the negative correlation in these experiments.

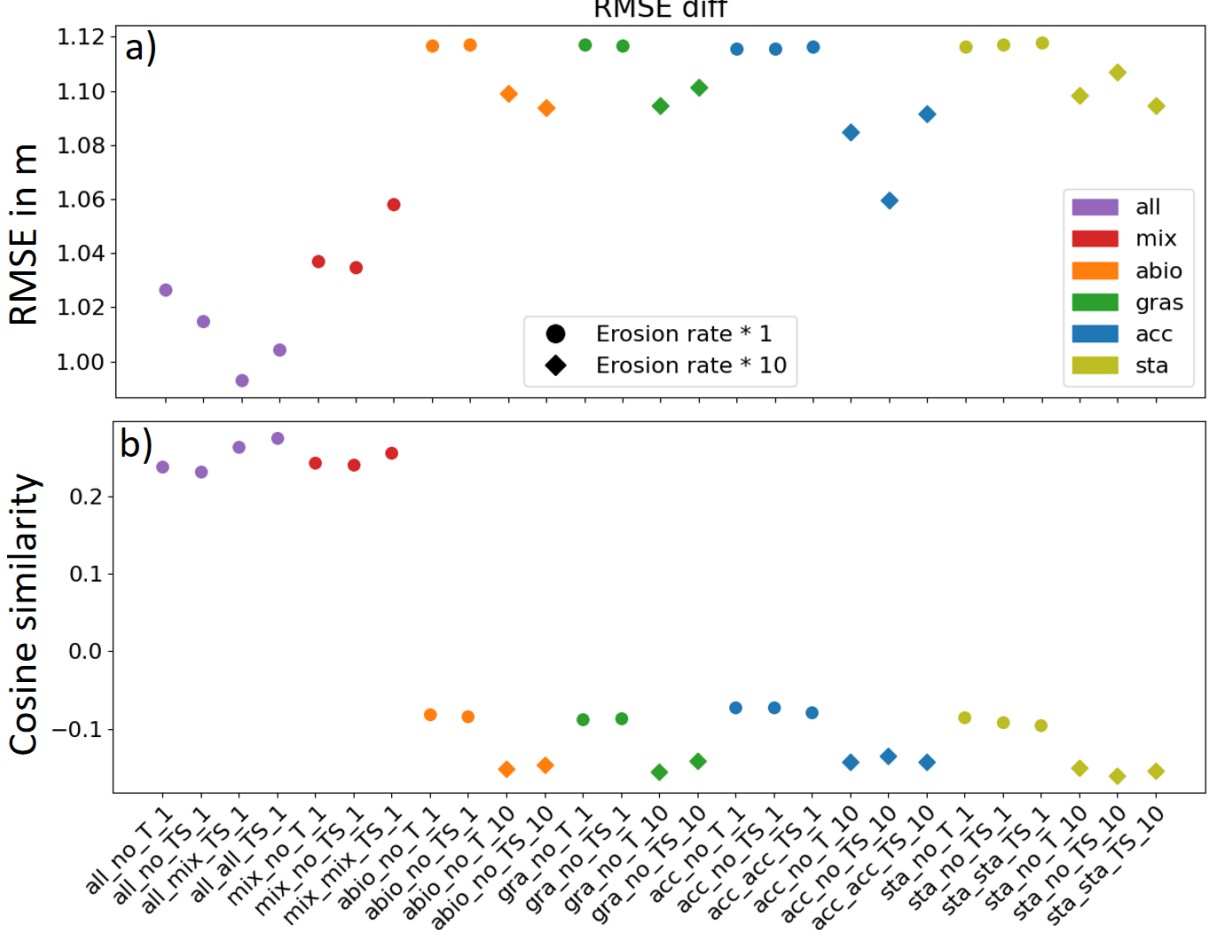

**Figure 4. Performance of all simulations in terms of (a) RMSE between the modeled and measured water depth change over the entire bay and (b) cosine similarity in the main channels. The values 1, -1 and 0 indicate positive, negative and no correlation between modeled and measured depth change, respectively. Diamond markers indicate the simulations in which erosion rates were increased by a factor of 10. From left to right, for each experiment with an individual**

**functional group, the model complexity is increased from a normal run without storms, to a run including storms, and lastly including seasonality of benthos effect (Table 3).**

### 4.3 Morphological development

The spatial difference in the model results among the experiments and comparison with the measurement is shown

in Fig. 5. Measured data indicate net deposition (up to 0.8 m) inside the main tidal channels accompanied by net erosion (up to 1.2 m) at adjacent flats from 2001 to 2009 (Fig. 5b, 6). Compared to a dominant deposition pattern in the channels, the tidal flats exhibit both erosion and deposition in large parts, including various bar-like structures mostly within the range of ±0.2 m. However, these structures are likely attributed to artifacts caused by measurement uncertainties and data processing which partly explain the discrepancy in the average depth of tidal

flats between measurement and model simulations (Fig. 5). Therefore we mainly focus on those apparent deposition and erosion patterns in the channels and adjacent flats that exceed the measurement uncertainties. As indicated in the cosine similarity analysis, only the experiments with all benthos (all_x) and with inclusion of only biomixers (bio_x) are able to reproduce the extensive deposition pattern in the tidal channels (Fig. 5b, Fig.6), whilst other experiments including those reference runs which consider only abiotic drivers show dominance of

erosion in the main channels (Fig. 5c&d, Fig.6). The reference run based on the original formulation of erosion rate (Pinto et al., 2012) produces morphological change within the range of ±0.1 m (Fig. 5c), which is much smaller than the measured values (Fig. 5a). Only by an increase of the erosion rate by a factor of 10 the reference run is able to produce morphological changes that are at the same order of magnitude with the measurement (Fig. 5d).

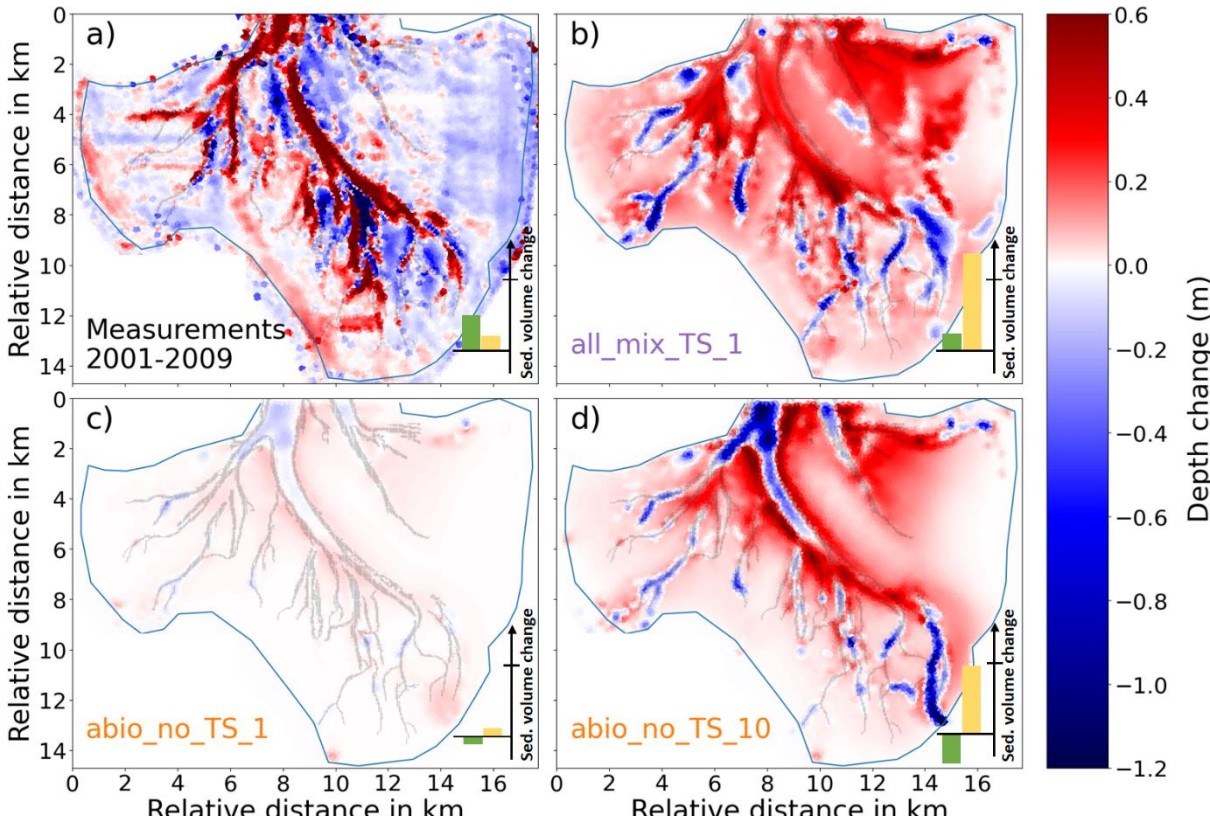

**Figure 5. Comparison of morphological change from 2001 to 2009 between the model experiments and the measurement: (a) result of All2; (b) measurement; (c) result of Ref1b; (d) result of Ref1. Positive and negative values are for deposition and erosion, respectively. The bars in the lower right corner represent the total sediment volume change in the main channel (green bar) and the basin excluding the channel (yellow bar). Negative/positive values indicate erosion/deposition. The mark on the y-axis indicates $10^7 \, \text{m}^3$. In the measured data, only the grid cells where the morphological change exceeds the measurement uncertainty (standard deviation of difference between the 2001 and 2009 field data) were included in the sediment budget analysis.**

There exists a net sediment input to the Jade Bay from 2001 to 2009 (~ $0.7 \cdot 10^7 \, \text{m}^3$), which is indicated by the measurement and captured by model experiments to various extent (Fig. 5). Increased sediment input into Jade Bay was also reported by Benninghoff and Winter (2019). However, most experiments overestimate the volumetric import compared to the measurement, especially on the tidal flats, and the magnitude varies among the experiments (Supplement material), with largest values in the runs which include the combined effect of all benthos Measurement data indicate that the net gain of sediment in the main channel exceeds the net import through the inlet of the bay, suggesting that the sediment accumulated in the channel originates not only from external sources outside the bay but also from internal sources, e.g. erosion at adjacent flats. Simulation results suggest that sands accumulated in the channels mainly come from internal sources whilst mud may originate from both internal and remote sources outside the bay (Fig. S4, Supplement material). Despite an overestimation of net sediment import to the bay, the model experiments with all benthos included (all_mix_TS_1) produce less deposition in the main channel compared to the measurement (Fig. 6). Instead, much of the imported sediment is deposited over an extensive part of the tidal flats in these runs, as exemplified in Fig. 5a. The reference experiments which include only abiotic drivers (abio_x) indicate little or none net sediment accumulation in the channel despite net sediment import through the inlet. In these runs, imported and eroded sediments from the main channel are mostly deposited along the edges of the channels on the flats (Fig. 5c &d).

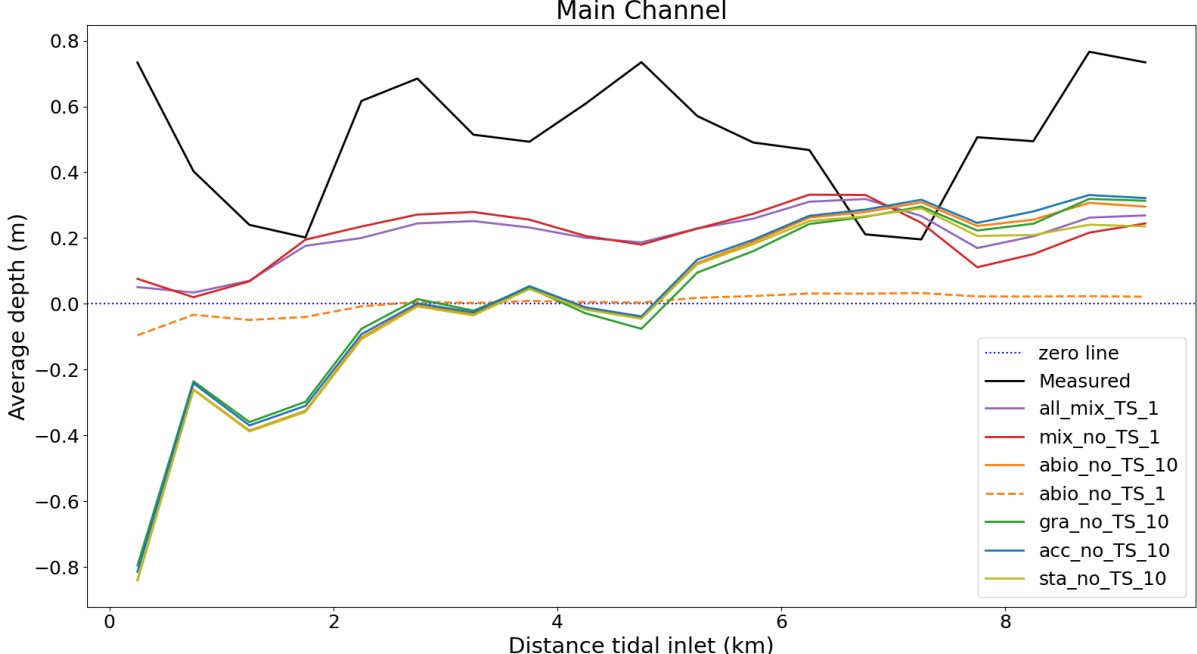

**Figure 6. Average depth change in the main channel calculated from the measured data and seven representative model experiments between 2001 and 2009. The 0 km in the x-axis marks the position of the inlet directed into the basin.**

### 4.4 Change in sediment composition

There exists remarkable changes in sediment composition in the Jade Bay from 1996 to 2009 according to Ritzmann and Baumberg (2013). Comparison between the observed change and model results indicate that the changes are largely reproduced in the experiments but no experiment alone captures all observed changes (Fig. 7). The best performance is shown in the experiments which include all benthos (all_x). Most of the large-scale changes in sediment composition (indicated by ellipses with roman number I-V) are satisfactorily reproduced in all_mix_TS_1, except for the area in the northwest part of the bay (I) where an opposite result is shown in the experiment (Fig. 7a&b&e). On the contrary, experiments which include only abiotic drivers are able to capture the observed change in this area (Fig. 7d&e), but with a worse performance in other areas. The experiment which includes only abiotic drivers and based on the original formulation of erosion rate (abio_no_TS_1) produces only an increase of mud content but fails to capture the loss of mud (Fig. 7c&e). Figure 7a illustrates changes in the flat type according to changes in mud content. Since the original mud content change data were not available, flat type change instead of mud content change was compared in this study, which restricts the comparison to a qualitative manner

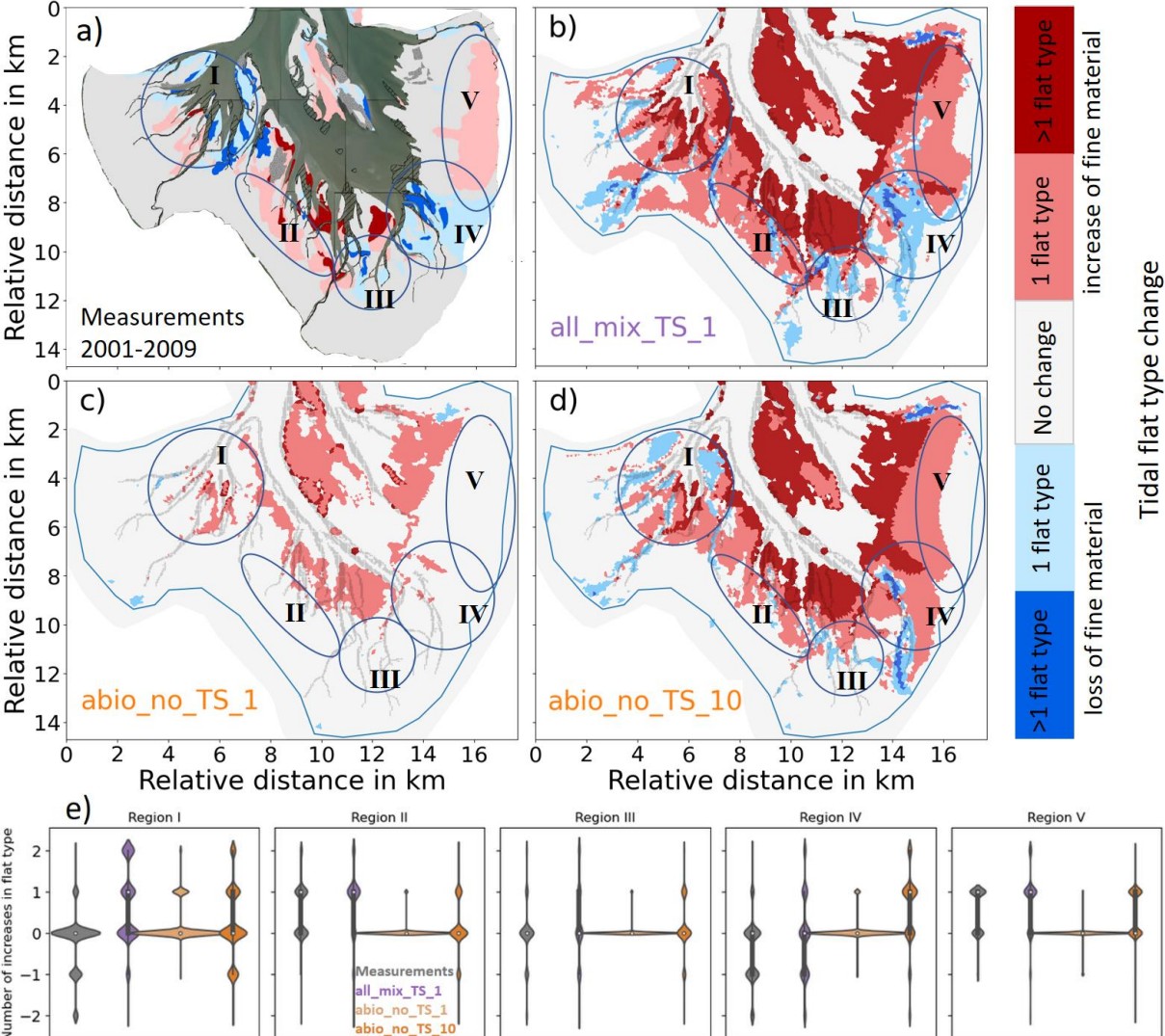

**Figure 7. Comparison of change in sediment composition between 2001 and 2009 between model results and observation: (a) result of all_mix_TS_1; (b) observation; (c) result of abio_no_TS_1; (d) result of abio_no_TS_10. Pale red and pale blue show the areas where the amount of fine sediment increased or decreased respectively with a change by one tidal flat type (according to Fig. 1b). Red and blue show areas with changes by two or more tidal flat types. Areas featured by large-scale changes are marked by ellipses. (a) shows a modified version of a plot from Ritzmann and Baumberg (2013) and was kindly provided by the NLWKN (Niedersächsischer Landesbetrieb für Wasserwirtschaft, Küsten- und Naturschutz). The dark grey area in (a) marks the area where Ritzmann and Baumberg (2013) could not obtain data due to permanent inundation. The roman numbers indicate areas to compare the measurements with the simulations. Subfigure e) shows the violin plot of the 5 denoted regions in a-d for each of the scenarios. The width of the violin shows the probability distribution and the white dot indicates the median.**

## 4.5 Impact of benthos

To further figure out how the four functional groups of benthos contribute to changes in morphology and sediment composition, we compared the results of the model experiments which include the impact of individual functional groups with the reference experiments which include only abiotic drivers. Since each group of experiments consists of several runs with different complexity (Table 3), we chose the run from each group with the least RMSE and same hydrodynamic conditions for comparison, namely abio_no_TS_10, mix_no_TS_1, acc_no_TS_10, gra_no_TS_10 and sta_no_TS_10.

### 4.5.1 Biomixers

The difference in the depth change between the runs with benthos and the reference run abio_no_TS_10 shows that the largest difference in the morphological change is caused by biomixers (Fig. 8a), followed by accumulators,

seagrass and stabilizers (Fig. 8b, c & d). In particular, the extensive accumulation of sediment in the main channel, which is shown in the measurement (Fig. 5a), is associated to the impact of biomixers. The impact of biomixers also causes deposition over a large part of the shallow tidal flats, as well as erosion at the flats adjacent to the tidal channels. The joint effect leads to a smoothing of the depth gradients between the channels and adjacent tidal flats. Morphological changes caused by biomixers are in the range of ±1 m compared to the reference run. It is worth noting that biomixers account for not only the enhanced deposition in the main channel, but also the decrease of mud content in the southern and southeastern parts (III and IV) of the bay (Fig. 9a&e). These changes are in consistency with field data.

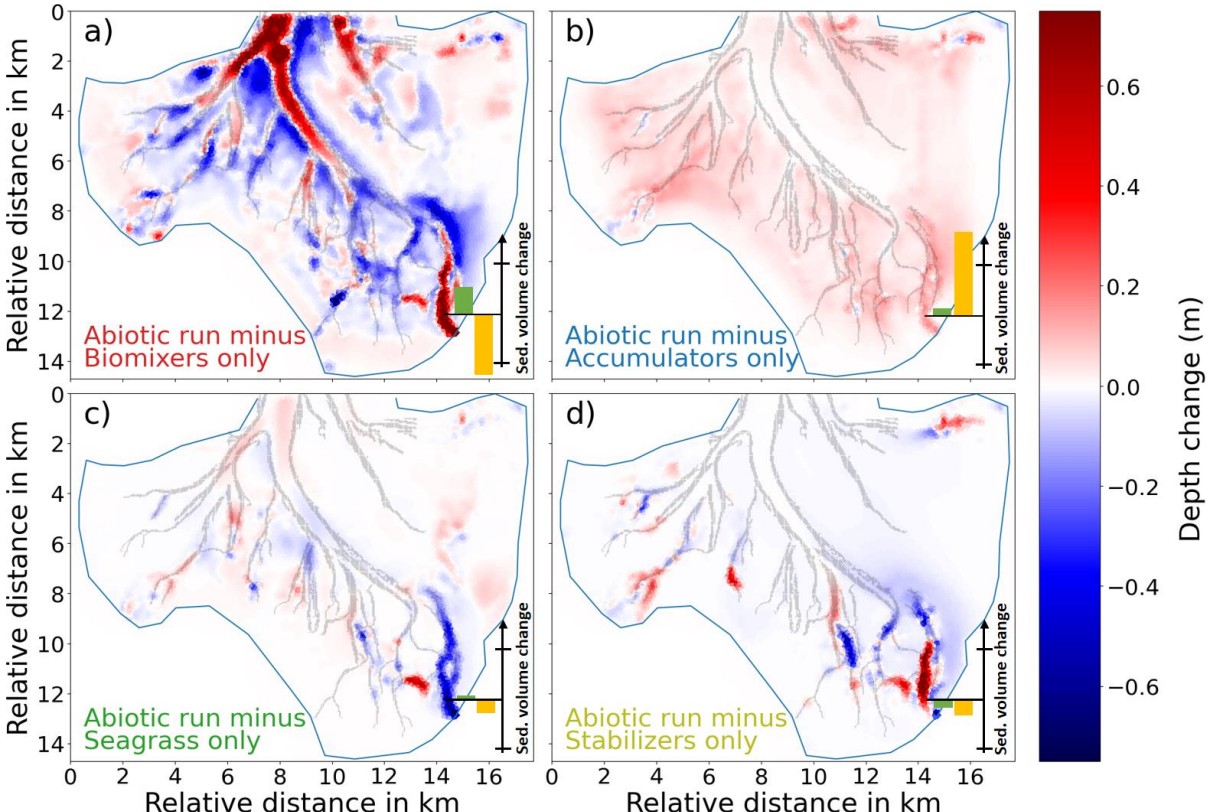

**Figure 8. Difference in the depth change between the reference run abio_no_TS_10 and (a) mix_no_TS_1, (b) acc_no_TS_10, (c) gra_no_TS_10 and (d) sta_no_TS_10. Positive and negative values indicate increased deposition and erosion, respectively, in the runs with benthos compared to the reference run. The bars in the lower right corner represent the total sediment volume change in the main channel (green bar) and the basin excluding the channel (yellow bar). Negative/positive values indicate erosion/deposition. The marks on the y-axis indicates $\pm 3 \cdot 10^6 \text{m}^3$.**

### 4.5.2 Accumulators

The presence of accumulators causes an overall enhanced deposition over a vast part of the tidal flats, with local values up to 0.5 m when compared to the reference run (Fig. 8b). The average deposition over at the tidal flats is highest compared to other simulations (Fig S6b). Accumulators do not seem to directly impact the morphological change of tidal channels, however, model results show that they can lead to a significant increase of mud content in a vast part of the bay including the channels (Fig. 9b&e). In particular, the observed increase of mud content in the southwestern part (II) of the bay is attributed to the impact of accumulators according to the model result.

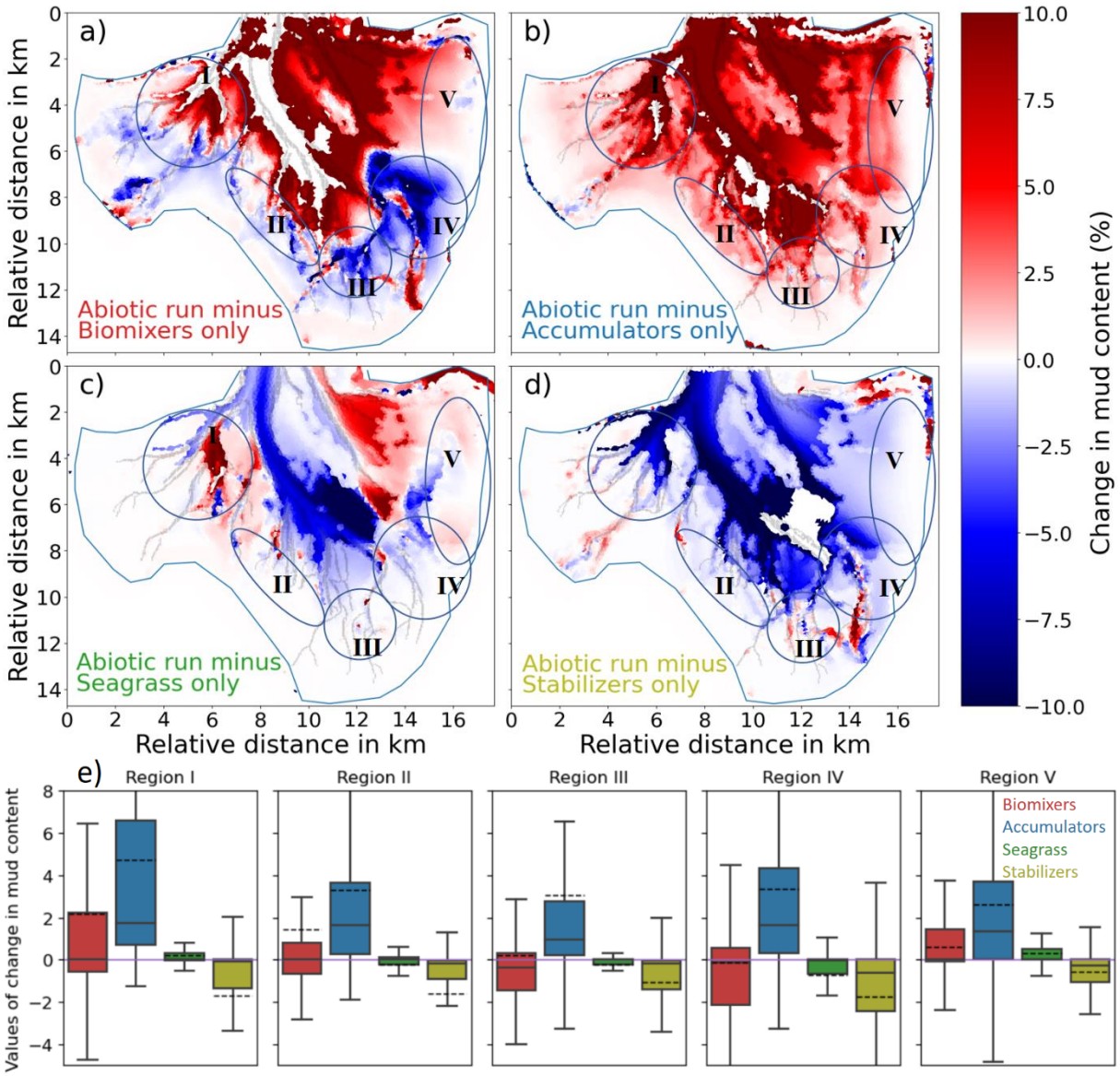

Figure 9. Difference in the mud content (%) between the reference run_abio_no_TS_10 and (a) mix_no_TS_1, (b) acc_no_TS_10, (c) gra_no_TS_10 and (d) sta_no_TS_10. Subfigure e) shows the boxplot diagram of the 5 denoted regions in a-d for each of the scenarios. The zero line is indicated in purple. The median (black solid line) and the mean (dashed black line) are shown in the boxplot.

### 4.5.3 Seagrass

Our simulation results suggest that the impact of seagrass on morphological change of the Jade Bay is smaller than that of biomixers and accumulators when looking at the overall depthchange (dark red and blue bars in Fig. 8). However, local changes might be higher compared to the accumulators scenarios (Fig. 8b, c). Further, instead of tidal flats, channels and areas adjacent to seagrass meadows are particularly under high impact. In the eastern part of the bay where seagrass is present, a slight deposition in the range of 20 cm occurs at the edge and outer parts of the seagrass meadows (Fig. 8c). Meanwhile, mud content decreases in the same area, suggesting a winnowing process there (Fig. 9c&e).

Interestingly, seagrass meadows affect not only sediment transport and morphodynamics in the direct vicinity around their habitats, but also causes far-reaching changes over the bay including the channels and other flats that are free of seagrass (Fig. 8c & 9c). This effect is through a feedback of seagrass meadows to larger-scale hydrodynamics. The ratio in the transported volume between the flooding and the ebbing phase calculated from

the simulation results indicates that the majority of water enters the Jade Bay through its main channels during flooding phase and leaves it over the tidal flats during the ebbing phase (Fig. S5a). The spillway on the tidal flats in the east part of the bay (V), where seagrass meadows are located, experiences larger flow friction due to the presence of seagrass (Fig. S5b). As a consequence, more water is transported through the main channel, eroding more fine-grained sediments compared to the abiotic scenario (Fig. S5c). Thus, the increased loss of fine grained sediment in the main channel (Fig. 9c&e) correlates significantly with the changed water flux in the main channel (Fig. S5c).

### 4.5.4 Stabilizers

The impact of stabilizers on the morphological changes in Jade Bay is comparable to that of seagrass in magnitude. The resultant morphological change is mostly local within the habitats of stabilizers and featured by both erosion and deposition (Fig. 8d). Sediment stabilization and consolidation in the areas where stabilizers exist lead to reduction of sediment sources for the distal ends of small channels, preventing mobilization of sediments in these parts. Compared to the abiotic run the sediment budget in the tidal flat is negative (Fig. 8d). This is attributed to stabilization of tidal flats outside of Jade Bay, leading to less erosion there and thus less sediment transport from outside into the Jade Bay. The impact of stabilizers on sediment composition is more prominent compared to the morphological change. In the subtidal area, a significant decrease of mud content is seen in the simulation result compared to the reference experiment (Fig. 9d&e), as a consequence of reduced mud input from stabilized areas, predominantly on the distant tidal flats.

## 5. Discussion

### 5.1 Model hindcast and implication

The model performance, both in terms of morphology and sediment distribution, are improved when biota is included in the simulation. In particular, the extensive deposition in the main channels is reproduced only by the experiments with either combined effect of all benthos (all_x) or with biomixers (mix_x), whilst other experiments produce an opposite pattern.

Our simulation results show that, among all four functional groups considered in the modeling, biomixers are most impactful on morphological change of the Jade Bay, followed by accumulators, seagrass and stabilizers. The morphological change of the bay over the 8.5-years period (2001-2009) is featured by distinct deposition inside the main channels and erosion at their adjacent flats (Fig. 5a). This feature and the amount of deposited sediment could be reproduced by modeling only when the impact of benthos, especially biomixers, is included.

The impact of biomixers on sediment is mainly destabilization (Arlinghaus et al., 2021) but can, under certain circumstances, exert stabilization as well (Cozzoli et al., 2019). This depends on metabolic rate, bottom shear stress and sediment composition. Muddy sediment particles in general have a higher organic matter content and therefore higher nutritional value than sands, and are hence more intensively reworked and bioturbated by benthic fauna (Arlinghaus et al., 2021). In sandy sediments, benthos-produced mucus exerts a stabilization impact which often exceed the destabilization impact because of less bioturbation (Orvain, 2002; Le Hir et al., 2007). For this reason, the channel deposition can be explained by two factors related to macrobenthos. Firstly, the critical shear stress for erosion is increased by the presence of biomixers ($p_d > 1$ in Equation 1; Fig S3) in the sandy channels, leading to enhanced resistance to erosion. Secondly, enhanced erosion on the tidal flats by biomixers ($p_d < 1$, $g_d > 1$) mobilizes sands which are partly deposited in the channel. Mud can hardly accumulate in the channel due to a low

sinking velocity and low threshold for resuspension (before consolidation). The majority of the accumulated sands in the channels comes from the eroded tidal flats. The redistribution of sediments from the tidal flats, which become increasingly deeper, into the channels, which become shallower, represents a typical basin development pattern under the impact of biotic destabilization as demonstrated by Arlinghaus et al. (2022). This is the case for the Jade Bay where a shift of functional groups took place between the 1970s and 2000s with biomixers increasing from ~ 20% to almost 70% in the field surveys (Schückel and Kröncke, 2013). Furthermore, the channel incision and sediment deposition at its edges in the model experiment which considers only abiotic drivers compare well with the abiotic scenario presented in Arlinghaus et al. (2022), in which deep and narrow channels develop with shallow tidal flats. The effect of unrealistically strong channel incision is known in morphodynamic modeling, although this problem is often overlooked (Baar et al., 2019). One practical solution that is often adopted in applications is an increase of the bed slope diffusion, e.g. by up to a factor of 100 (Van der Wegen and Roelvink, 2012; Schuurman et al., 2013; Braat et al., 2017). However, this solution does not represent a process-based understanding. An alternative solution is provided in our modeling study which proposes to include the impact of bioturbation instead of tuning the bed slope diffusion.

Compared to the remarkable impact of biomixers which leads to deposition in the channels and erosion in the flats and therefore a general widening of channels, other functional groups have less influence in the morphological change of the main channels according to our simulation results. Accumulators mainly enhance sediment deposition on the tidal flats. Seagrass meadows can modify the flows not only within or adjacent to their habitats but also at a large-scale covering a vast part of the bay, which results in alternating erosion and deposition patterns in the main channel. The impact of stabilizers on the morphological change of the Jade Bay is small compared to biomixers and accumulators. This is attributed to their location. The shallow tidal flats in the south and west of Jade Bay which are inhabited by stabilizers are subject to relatively weak tidal currents and low SSC. The different impacts of the mentioned functional groups in the Jade Bay are depicted in simplified form in Figure 10 where sediment redistribution (e.g. from tidal flats to channels) and vertical erosion/deposition patterns are distinguished. Our results suggest benthos as a critical driver determining sediment stability and morphological development of tidal embayments and basins, supporting an earlier study by Backer et al. (2010). A reference simulation, which considers only abiotic drivers and adopts formulation of erosion rates from laboratory experiments in which benthos is excluded, heavily underestimates the morphological change. An increase of the erosion rate by a factor of 10 allows the reference simulation to produce morphological changes that are at the same order of magnitude with the measurement, but still fails to capture the spatial pattern. This indicates that existing formulations for sediment resuspension rate that do not take into account benthos impact may be of limited use for application to real coastal systems that are inhabited by benthos.

As demonstrated in the model results, the major effect of benthos is sediment mobilization and redistribution, which was also found in Borsje et al. (2008) and Lumborg et al. (2006). Especially import of mud into the bay is increased under the impacts of benthos, which is in line with other modeling results summarized in Arlinghaus et al. (2021). Our results show that accumulators have the strongest impact on changes in sediment composition, followed by biomixers, seagrass and stabilizers. The impact of accumulators is mostly local, but this functional group is present over a vast part of the bay and thus jointly leads to a large-scale impact. By contrast, the impact of biomixers extends beyond their habitats. Locally, sediment can be either stabilized or destabilized depending on the abundance of biomixers and other factors elucidated previously. Non-locally, the enhanced erosion in large parts of the tidal flats by biomixers increases the overall concentration of suspended sediment, especially on the

flats outside the Jade Bay, which provides a sediment source for the bay. The impact of seagrass is prominent in close vicinity to the meadows but not so much within the meadow itself. One explanation is that the effect of organic sediment accumulation due to primary and detritus production and root and rhizome formation, which are main sources for sediment production (Garcia et al., 2003), was not considered in this study. The found changes close to the meadows are in line with measurements indicating differences in bed level elevation between vegetated and non vegetated areas in the range of 3 cm per year (Potouroglou et al., 2017). The impact of seagrass meadows also reaches beyond their habitats by altering the large-scale hydrodynamics and the ratio of the inflow to the outflow in the tidal channels and on the flats. The increased loss of mud content in the tidal channels in the stabilizers experiments compared to the reference run can be explained by reduced supply of mud from the tidal flats which are inhabited by stabilizers. However, since mud content is small in the hydrodynamically active areas, the absolute change of mud content induced by stabilizers is minor.

The changes in sediment composition are reproduced more satisfactorily in four areas with the inclusion of benthos effects, namely the southern (III), the southeastern (IV), the eastern (V) and the southwestern (II) parts of the bay (Fig. 7). The loss of mud due to erosion in the southern (III) and the southeastern (IV) parts is mostly attributed to the impact of biomixers which has a strong destabilization effect there. The eastern (V) part accumulates much more fine sediment compared to the reference run, which is attributed to the impact of seagrass and accumulators (Fig. 9). This impact might even be enhanced in reality due to the organic sediment accumulation explained above. The increase of mud content on the shallow tidal flats in the southwestern part is mainly due to the presence of accumulators. At one site in the western part, the reference simulation yields better results with a loss of mud, which is not captured by experiments with benthos.

Overall, the increase of mud content is overestimated in all model experiments when compared to the field data. One possible explanation is that mixing between sediment layers, which gets enhanced by biomixers, was not implemented in the model and thus all freshly deposited mud remains on the seabed surface before being eroded at a later stage or buried by further new deposits, whilst mixing in the sediment column in a natural system would mix freshly deposited mud and organic matter with other coarser particles and lead to homogenization of sediment grain size in the upper 10-30 cm as pointed out by previous studies (Knaapen et al., 2003; Paarlberg et al., 2005; Arlinghaus et al., 2022).

It should be noted that the dominant impact of biomixers and accumulators is related to their widespread abundance and high biomass in the Jade Bay. In other environments, different functional groups may dominate. For instance, some modeling studies show a significant impact of seagrass on morphodynamics of tidal basins (Mohr, 2022), barrier islands (Reeves et al., 2020) and estuaries (Walter et al., 2020). Seagrass impact may further complicate when their effect interacts with other plants such as salt marshes (Carr et al., 2018). Unfortunately, a quantitative comparison of impact normalized to biomass between the different functional groups cannot be made in this study due to lack of biomass data of seagrass and stabilizers in the study area, which points out a need for future studies.

**5.2 Societal relevance**

Similar to many other coastal bays/embayments worldwide, the Jade Bay serves important socio-economic functions for tourism, logistics and on the other hand provides important refuge for a variety of marine lifeforms. It is of critical importance to sustain the ecological functions of coastal bays such as the Jade Bay under the increasing pressure of human use and climate change. Our results indicate that benthos can significantly modify morphological change and sediment composition in tidal embayments, and can play a key role in the natural resilience of coastal systems against human and climate stressors. However, we also revealed that the impact on

morphological development varies among different functional groups. Biomixers tend to smooth the bathymetric gradients between channels and flats, whilst seagrass and accumulators may counteract this to various extents. A combined effect of all functional groups leads to increased import of sediment, especially mud, to the bay. Our results support the hypothesis by Haas et al. (2018), who proposed that an abundance of mud and eco-engineering species often culminates in continuous embayment filling with fine sediment and the growth of intertidal and supratidal areas, eventually leading to closure of the embayment. However, on the other hand, there is growing concern about whether coastal systems such as the Wadden Sea including the Jade Bay can keep pace with the foreseeable sea level rise for the upcoming decades (Plater and Kirby, 2011). Our results show that the morphological development of the Jade Bay is able to sustain the impact of sea level rise, at least for the period 2001-2009, because of a net sediment import caused by a joint effect of abiotic and biotic drivers. But it is unclear how the drivers would change in future, especially how the different functional groups of benthos would react to human and climate stressors. For instance, chlorine inputs are expected to increase in the Jade Bay due to construction of liquefied natural gas (LNG) terminals, which will likely have an impact on the population, abundance and distribution of the different functional groups. This may result in a loss of sensitive species and functional groups as pointed out by studies in other regions (Chang, 1989; Wang et al., 2022). Extreme weather events, such as heat waves, also have a significant impact on benthos (Serrano et al., 2021). Intensity and frequency of extreme events are likely to increase in the future due to climate change, but the consequent change in benthos remains largely unknown. To this end, explanatory and eventually predictive numerical models are imperative for exploring feasible nature-based solutions for sustaining both socio-economic and ecological functions of coastal regions.

**5.3 Model limitations and future research needs**

Earth system modeling and regional modeling inevitably comprise uncertainties, which originate from various sources including boundary conditions, numerical solvers, and parameterization of processes. This is especially true in modeling of coastal systems in which physical and biological factors may be of comparable importance in guiding the system evolution. Model refinement and/or inclusion of additional processes do not necessarily increase model accuracy since the uncertainties in parametrization of less-known processes (e.g. growth/decline of benthos, interactions between different species/functional groups) may exceed the gain in accuracy (Skinner et al., 2018, Pianosi et al., 2016). An earlier study found that it remains a challenge to get physically correct results for both sediment transport and morphodynamics simultaneously (Baar et al., 2019). Therefore, development of hydro-eco-morphodynamic models will always be limited to a certain tradeoff between complexity and accuracy. This is confirmed in our study, which indicates that an increase in model complexity by considering the benthos impact firstly increases model performance in approximating observed change, but model performance decreases when a higher complexity, i.e. seasonal change of benthos, is added by a simple parameterization. This points out a need for an accurate mapping of benthos including their temporal changes in field which can serve input for the modeling and/or process-based understanding and formulation of temporal change of benthos for modeling.

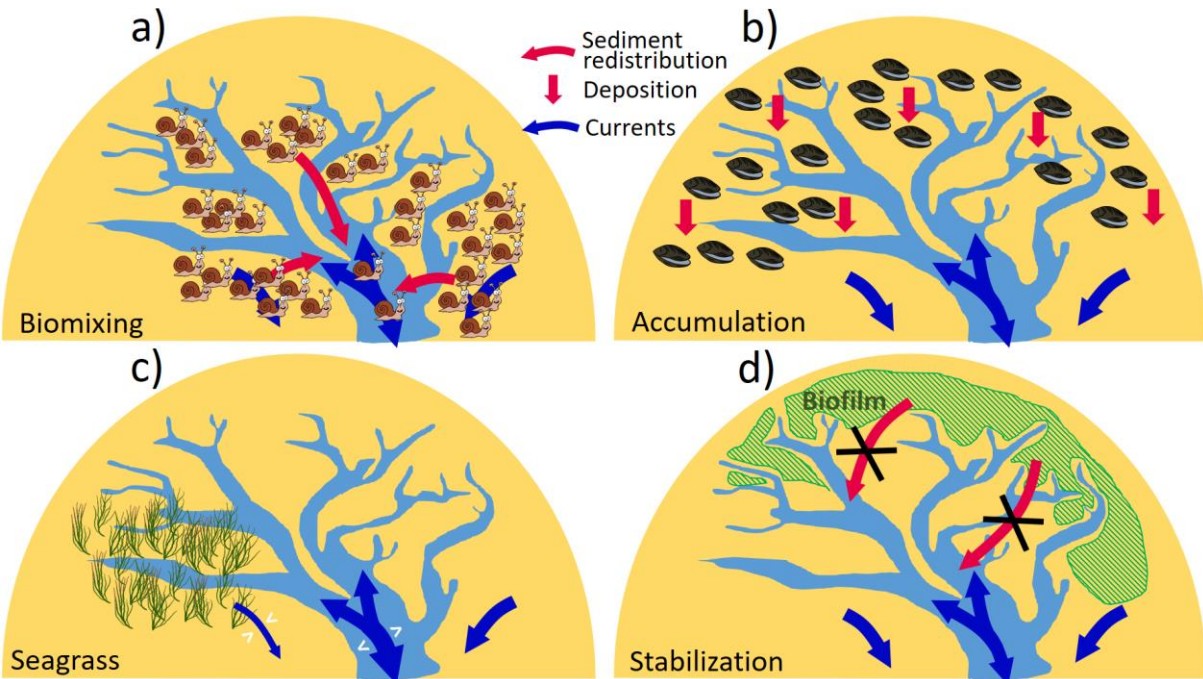

**Figure 10. Conceptual sketch of different effects of the four functional groups on sedimentation and hydrodynamics in tidal embayments: (a) destabilization in tidal flats caused by biomixers, (b) accumulation caused by filter / suspension feeders, (c) modification of flooding / ebbing flows by seagrass meadows, and (d) sediment stabilization by MPB and reduced input to channels.**

## 6. Conclusions

We have presented an effort towards large-scale explanatory hydro-eco-morphodynamic modeling to explain changes in both morphology and sediment composition observed in a real coastal system, thereby disentangling the impacts of biotic and abiotic drivers. The following conclusions are drawn from the study:

Benthos significantly reworks sediment, thereby mediating large-scale and long-term change of coastal morphology and seabed sediment properties well beyond their habitats. Compared to the scenarios which include only abiotic drivers, simulations with benthos included produced significantly improved results that are closer to observation, and are able to explain some unique features in the historical change of morphology and sediment composition in the Jade Bay. The most impactful functional group regarding morphological change in the Jade Bay is biomixers. The impact of biomixers leads to prominent sediment accumulation in the main channels. Accumulators mainly enhance sediment deposition on the tidal flats. Seagrass meadows modify the flows not only within or adjacent to the sites where they are located but also at a much larger scale beyond their habitats, resulting in alternating erosion and deposition patterns in the main channels. Stabilizers locally prevent mobilization of sediments on the distant tidal flats. Regarding the change of sediment composition in the Jade Bay, accumulators have the strongest impact. The impact of accumulators is mostly local, but this functional group is present over a vast part of the bay and thus jointly leads to a large-scale impact. By contrast, the impact of biomixers, seagrass and stabilizers on sediment composition extends beyond their habitats. A combined effect of all functional groups leads to increased import of sediment, especially mud, to the bay. Also, results indicate that impacts of functional groups can both counteract and enhance each other. Increased SSC level by biomixers for instance, enhances the impact of other functional groups. On the other hand, biomixing-induced sediment erosion on the tidal flats is partly offset by the bio-deposition of accumulators.

Our results further show that increasing model complexity does not necessarily lead to better model performance, especially when biotic drivers such as benthos is included. Including storm surges, which are precisely described by observational data, improves model performance. By contrast, adding seasonality in benthos impact through oversimplified parameterization decreases the general model performance. The reason is attributed to lack of observational data which can support a more accurate formulation of temporal changes of benthos behaviors. Therefore, the complexity of hydro-eco-morphodynamic models should be balanced at a certain level on which a tradeoff between complexity and accuracy can be obtained.

Coastal systems such as the Jade Bay have important socio-economic and ecological functions worldwide. Therefore, development of advanced numerical models which are able to explain and predict the states of coastal morphology and sediment properties and to develop measures for protection is of vital importance. To achieve this step, further effort in numerical modeling should explicitly include biotic drivers such as benthos and deepen the understanding on the interactions between different functional groups and between biota and abiotic drivers. In this sense, not only dedicated field measurements and lab experiments but also large-scale and long-term monitoring are indispensable.

### Acknowledgements

This study is a contribution to the I$^2$B project "Unravelling the linkages between benthic biological functioning, biogeochemistry and coastal morphodynamics – from big data to mechanistic modelling" funded by Helmholtz-Zentrum Hereon. It is also supported by the Helmholtz PoF programme "The Changing Earth – Sustaining our Future" on its Topic 4: Coastal zones at a time of global change.

The benthic infauna dataset form 2009 was kindly provided by the Senckenberg am Meer. The sediment map was provided by the NLWKN (Niedersächsischer Landesbetrieb für Wasserwirtschaft, Küsten- und Naturschutz).

### Authors contribution

Peter Arlinghaus designed the study and performed numerical simulations. Wenyan Zhang designed and supervised the study. Peter Arlinghaus and Wenyan Zhang wrote the manuscript. Ingrid Kröncke provided the infauna dataset. All four authors were involved in manuscript revision.

### Data statement

Publicly available datasets were analyzed in this study. Morphological data of the German Bight can be found at: https://datenrepository.baw.de/startseite. Sea level data at tide gauge station Wilhelmshaven can be found at: https://map.emodnet-physics.eu/platformpage/?platformid=9044&source=cp.

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
