# Peer review of "Benthos as a key driver of morphological change in coastal regions"

_EGUsphere, 2023_

## Author Response (AR1)

**Reply on RC1**

We thank Referee #1 for the constructive comments that have helped us to clarify and improve our manuscript. Our responses to the specific questions/requests (in **bold & *italic***) are listed below.

**Assessment:**
***(…)Mostly my confusion was in the description of the different model runs (i.e., Table 3) and how the description of storms (which seemed to have significant impact) were not explained in detail. (…)***

Thank you for pointing this out. We have improved the description in naming the functional groups and parameter settings in the associated sensitivity model runs.

Abbreviations for the functional groups, seasonality, hydrodynamic forcing and sediment parameter

are:

Biomixers = mix

Accumulators = acc

Stabilizers = sta

Seagrass = gra

Inclusion of all four functional groups = all

Abiotic model run without consideration of any benthos effect = abio

Seasonal variation of benthos excluded/included = no / abbreviation of specific functional group(s)

Hydrodynamic forcing excluding/including storm surges = T / TS

Erosion rate by default / scaled by factor of 10 = 1 / 10

The experiments are named by combination of the different model features separated by an underscore and read as:

**Modeled functional groups_Seasonality_Hydrodynamics_Erosion Rate**

A full set of experiments with the modified names is provided in the updated Table 3, including

 all_no_T_1
 all_no_TS_1
 all_mix_TS_1
 all_all_TS_1
 mix_no_T_1
 mix_no_TS_1
 mix_mix_TS_1
 sta_no_T_1
 sta_no_TS_1
 sta_sta_TS_1
 sta_no_T_10
 sta_no_TS_10
 sta_sta_TS_10

acc_no_T_1
    acc_no_TS_1
    acc_acc_TS_1
    acc_no_T_10
    acc_no_TS_10
    acc_acc_Ts_10
    gra_no_T_1
    gra_no_TS_1
    gra_no_T_10
    gra_no_TS_10
    abio_no_T_1
    abio_no_TS_1
    abio_no_T_10
    abio_no_TS_10

A short description on how storms were implemented was shown in the supplementary material. We will expand this paragraph with a more detailed description.

**Specific comments:**

Line 23: We agree with the reviewer. This line will be changed in order to emphasize that "opposite" refers to morphological changes in terms of erosion and deposition.

Lines 99 101: We will change ca. to approximately to prevent confusion.

Line 137: Indeed, some of the parameters were determined from model simulations, that did not contain benthos. However, there is no conflict in using those parameters for predicting a current species distribution. The reviewer specifically questions if *mud content* and *shear stress* can be determined without benthos. The term *shear stress* should not be mistaken for *critical shear stress* for erosion. The latter is indeed strongly impacted by benthos, but the former is primarily depending on the hydrodynamics at the sediment-water interface. For the prediction of species distribution, *bottom shear stress* but not *critical shear stress* for erosion were used.
The applied mud contents are based on measurements (not simulations) and was used as a proxy for estimating species distribution. It is correct that benthos strongly impacts local mud content (and vice versa), but the induced small-scale changes may accumulate on time scales of years to decades to generate a large-scale effect. We agree that a more adaptive species distribution, which includes the mutual and dynamic feedback between mud content and benthos would be more accurate. This was however, out of scope in this study. We assume that other environmental parameters which were based on simulations and used for the estimation of species distribution such as inundation time or salinity are not significantly influenced by benthos.

Line 209: Instead of "vegetation proof", we will use the term "vegetation cover"

Table. 3: Please see our response to comment 1.

Lines 256-258: Actually storms were mentioned earlier in chapter **Model setup for the study area** in line 230. However we do understand why it was easy for the reviewer to overlook this point. In line 230 we only mention storms once and point to the supplementary. We will add another brief explanation here on how storms are implemented and extend the description in the supplementary.

Lines 306-320: We agree and will add a remark earlier in the text.

Lines 321-328: We agree and will update the abstract accordingly.

Figure 6b. This is true, caption and axis are inconsistent. We will improve this figure to provide more precise and quantitative information. The caption inconsistency will also be corrected.

Figure 7b: Plot 7b is not our work, but was adopted from Ritzmann and Baumberg (2009) with permission (as indicated in the caption). This plot is based on measurements and the "greenish grey area" indicates the area where no measurements were made. We will add this information to the caption.

Figure 7: One of the main purposes of this study is to show that simulation results are significantly improved if benthos impact is added. In order to prove this, we used historical morphological data, and data indicating the sediment change for model assessment. The sediment change data are shown in figure 7b. These data and this plot were not created by us, but adopted from Ritzmann and Baumberg (2009) with permission. The plot depicts the sediment change in terms of flat type change. Unfortunately we were not able to get the raw data that were used to produce the plot. We agree with the reviewer, that it would be more useful to compare composition changes and temporal transitions maps from one flat type to another. Because of lack of field data we had to stick to the flat type change comparison.
In order to avoid confusion here, we will provide explanation in the text on why flat type comparison was chosen, rather than actual composition change.

**Technical Edits:**

Line 129: Will be changed as suggested by the reviewer.

Figure 7 caption: NLWKN ("Niedersächsischer Landesbetrieb für Wasserwirtschaft, Küsten- und Naturschutz") is state agency that is subordinate to the German state administration. We will explain this in the text.

Line 500: Will be changed as suggested by the reviewer.

**Reply on RC2**

We thank Referee #2 for the constructive comments that have helped us to clarify and improve our manuscript. Our responses to the specific questions/requests (in **bold & *italic***) are listed below.

1. ***"(…) the naming conventions (as those presented in Table 3) are confusing for a casual reader."***

Thank you for pointing this out. We have improved the description in naming the functional groups and parameter settings in the associated sensitivity model runs.

Abbreviations for the functional groups, seasonality, hydrodynamic forcing and sediment parameter

are:

Biomixers = mix

Accumulators = acc3

Stabilizers = sta

Seagrass = gra

Inclusion of all four functional groups = all

Abiotic model run without consideration of any benthos effect = abio

Seasonal variation of benthos excluded/included = no / abbreviation of specific functional group(s)

Hydrodynamic forcing excluding/including storm surges = T / TS

Hydrodynamic erosion rate by default / increased by factor of 10 = 1 / 10

The experiments are named by combination of the different model features separated by an underscore and read as:

**Modeled functional groups_Seasonality_Hydrodynamics_Erosion Rate**

A full set of experiments with the modified names is provided in the updated Table 3, including

        all_no_T_1
        all_no_TS_1
        all_mix_TS_1
        all_all_TS_1
        mix_no_T_1
        mix_no_TS_1
        mix_mix_TS_1
        sta_no_T_1
        sta_no_TS_1
        sta_sta_TS_1
        sta_no_T_10
        sta_no_TS_10
        sta_sta_TS_10
        acc_no_T_1
        acc_no_TS_1

```
acc_acc_TS_1
acc_no_T_10
acc_no_TS_10
acc_acc_Ts_10
gra_no_T_1
gra_no_TS_1
gra_no_T_10
gra_no_TS_10
abio_no_T_1
abio_no_TS_1
abio_no_T_10
abio_no_TS_10
```

A short description on how storms were implemented was shown in the supplementary material. We will expand this paragraph with a more detailed description.

2. *"While you chose the model runs to optimize RMSE, it is not possible to evaluate the effects of the different biological groups if the baseline hydrodynamics are not the same."*

We agree, that the baseline hydrodynamics should be the same. This will be updated in the plots in the revised manuscript. Nevertheless, the main message and interpretations are not affected.

3. *"However, I worry that readers (who would cite your paper) may use your results to say that in general seagrasses don't affect morphodynamics, which is not always true."*

Indeed. This is a misleading message which our study would like to avoid. Despite that seagrass impact is less striking when compared to the major impact by bioturbators in the Jade Bay, seagrass meadows are indeed impactful and can change the morphology by several centimeters and up to tens of cm within a few years. More important is that seagrass meadows can modify the hydrodynamics and thus morphodynamics at scales well beyond their habitat. The discussion will be updated in this regard, highlighting that the differences in impact of different functional groups is strongly depending on their distribution and abundance/biomass, which can vary strongly among different sites. We will also add more references in the discussion highlighting the impact of seagrass.

4. *"Need additional results figures besides just maps. The maps only qualitatively show differences, no way for reader to quantitatively compare."*

We will add quantitative comparisons on the benthos effect in tidal channels and tidal flats. The newly added figure will show the average depth change in the channel as function of the distance from the tidal inlet for different biotic and abiotic runs and the depth change at the tidal flats as function of distance from the tidal inlet. It will be clear from this figure that, according to the measurements, the main channel have accumulated abundant sediment during the investigated time period, which was reproduced by the simulations including biomixers while other simulations failed.

The new figure and interpretation will be added to the main text.

The other two suggestions from the reviewer to create new maps were not realized for the following

reasons:

A graph with the distance that effects from benthos extends across domains is in principal possible to create. In case of the seagrass this can be done since it is very locally distributed. For other species however, this is not feasible since they are spread over the whole domain. And only considering areas with high biomass may lead to wrong results since benthos impact in region of small benthic biomass might be higher than in regions of high biomass depending on hydrodynamics (Cozzoli, 2016).

The third suggestion is about comparison between distal and proximal effects. This suggestion is similar to the previous one. If we understand the reviewer right, he/she asks for normalization of benthos impact of morphology, normalized by the metabolic rate of the different species. This is indeed a very interesting suggestion, which however we can unfortunately not realize. The reason is that for stabilizers and for seagrass no metabolic rates are known. Their abundance relies on rough estimates for stabilizers (Widdows and Brinsley, 2002; Daggers et al., 2020) and 2D abundance maps for seagrass (Adolph, 2010). Generating estimates for the metabolic rate are possible but will come with huge uncertainties. Furthermore it is not clear if the impact of macrophytes and micropythobenthos scales similar to the relation between microbenthic fauna and their metabolic rate since they are autotrophs and partially immobile.

5. *"Please elaborate on why you made certain modeling decisions by either citing more literature or discuss the decisions. (…) Additionally, did SSC remain constant for the study period (in real life)? (…) Additionally, do you assume that benthos abundance numbers remain constant? (…)"*

One of the ideas of this paper is to test the impact of different biological, hydrodynamic and sedimentological properties on the model outcomes in order to estimate the sensitivity of the morphology. For this reason major benthic functional groups including seasonal changes, storm impacts and sediment properties, namely the erosion rate, were investigated. In setups that included bioturbators (biomixers) the suspended sediment concentration (SSC) is strongly increased. Higher SSC makes the effect of filter feeders more pronounced since there is more material to deposit. Thus, making simulations with only accumulators is not sufficient to disentangle their impact in a realistic (all benthos) scenario, since their impact will be much weaker compared to those in a combined simulation of species. Increasing the hydrodynamic erosion rate by a factor of 10 aims to evaluate the effect that accumulators are having in the all benthos run.

The SSC coming from the open boundary is constant in the model. Likely there will be seasonal variability in SSC which we did not implement due to the lack of measurement data. To our knowledge there is no existing dataset showing turbidity values over longer time spans and on larger spatial scales in Jade Bay. The measurements we were able to find typically cover one or a few points measured over one or a few tidal cycles (Götschenberg and Kahlfeld, 2008; Becker, 2011). Measured values are in the same range as the one chosen in the paper (Becker, 2011). Other modeling studies from the Jade Bay show comparable amounts of suspended sediment as found in our simulation of Jade Bay (Kahlfeld and Schüttrumpf, 2006). This will be emphasized by putting a plot into the supplementary, showing the average SSC in the Jade Bay. Explanations and mentioned citations will be added to the main text.

There is unfortunately no field measurements on the inter-annual variability of benthos in the study area. We adopted the distribution derived from 2009 measurement. Seasonal variation was implemented as a simple function in some of the experiments. Other experiments assumed a stationary distribution. As pointed out in Arlinghaus et al. (2021) and Arlinghaus (2023, PhD thesis in press), the lack of species abundance/biomass distribution data is one of the major problems

hindering the development of large-scale morphodynamic modeling. This can/should be improved by implementing regular monitoring schemes. We would like to emphasize here that, although the field dataset lacks a temporal evolution, it represents the most comprehensive dataset from the Wadden sea that is used in a large-scale bio-morphodynamic modeling study, see Table 2 in Arlinghaus et al. (2021).

We will add discussion of each point above in the main text.

*6. Consistency in naming. Please improve the consistency with how you refer to benthos functional groups/species. (…)*
We agree. This has been clarified in the response to the first major comment. Please see above.

*7. You could consider calculating the RMSE for areas with high biomass of the given species (for example seagrass) to get at the local effects and normalize for the fact that seagrass is only found in a small area compared to more extensive presence of bioturbators.*
This comment is similar to an earlier one (comment 4). Unfortunately, as explained above, the normalization regarding biomass of metabolic rate for seagrass and other functional groups is not feasible with the data that we have due to large uncertainty.

*8. Strengthen answer to research question 2. As is written now, your rational for choosing the model runs you show is not compelling.*
We agree that especially the question for combined impacts of benthos is not elaborated thoroughly enough. We will do so in the text.

*9. Can you isolate the individual contributions of each functional group within All2 run? (…)*
As assumed by the reviewer this is only partially possible since the functional groups interact and thus a single functional group simulation can never completely represent the effect that this functional group would have when presented with other functional groups. However, one way interaction is mediated between functional groups, which is due to the available amount of suspended sediment. To reach a comparable level of suspended sediment to the scenarios in which the sediment is strongly destabilized by bioturbators, the hydrodynamic erosion rate was artificially increased by a factor of 10. A respective note will be added to the text.

*10. "Also the approach you take is slanting your results towards bioturbators because they are the most abundant. Can you normalize by downsizing to a subdomain (…)"*
As explained in several comments above this is not feasible since biomass values for seagrass and stabilizers are not known. To address this we will add a discussion in the main text, pointing out the need to further investigate the relative impact of species/functional groups, compared to its biomass and metabolic rate.

*11. "If unable do 1 & 2 based on model limitations, choose same hydrodynamic conditions. (…)"*
In context of suggestion 1 it is valid to use RMSE to compare the functional groups, even though they are presented in different abundances and biomasses. Because it shows, which functional group contributes most to the changes observed in the study area.
If we understood the reviewer right, he/she wants to get answer for the following question:
"Assuming the same biomass and/or metabolic rate and the same spatial distribution in a patch that experiences the same hydrodynamic conditions, which functional group will have the largest impact on the morphology? If this cannot be answered, cross out biomass and metabolic rate from this

question".

As mentioned above, answering this question offers sufficient material to make a new study by itself. For example, to answer this question we would need to make a new model simulation, choosing a small patch where all four functional groups are abundant and on the other hand remove all benthos outside this patch (since benthos effect reaches well beyond their habitats). We also need to be careful when choosing the patch, since every part of the bay has a unique impact on the overall hydro-morphodynamics of the whole system. We believe that answering this question is very interesting and worthwhile, but would go beyond the scope of this study and lead to a very lengthy but not well streamlined paper. We hope the reviewer would agree with our argument.

**Figures:**

In general:
It is not uncommon to use blue color for erosion and red for accumulation of sediments as can be seen in Benninghoff et al., (2019) or Brückner et al., (2021). For this reason we prefer to keep the color presentation.
In the choice of the colormap we are limited to a divergent colormap. In order to better distinguish the subtle changes, *seismic* instead of *bwr* colormap will be used in an updated plot, which has higher saturation in color. Additionally the range for the depth change will be adapted in order to emphasize small changes. The information lost, when cropping the colorbar will be compensated by the new picture suggested by the reviewer and that will be added to the updated version of the paper. It shows the actual depth change in the channel and tidal flats.

Figure 1:
We agree: Scale bar and North arrow will be added to the plot.

Figure 2:
Both common names and cartoons will be added to the figures. That the biomass is constant will be mentioned in the text.

Figure 3:
We will keep the Taylor diagram. One of the circles will be changed to a triangle as suggested.

Figure 4:
This is a good idea. We will color all model run names that appear in figures.

Figure 5:
The figures will be updated according to the new names. Also increased erosion rate is indicated in the new model run names.
As can be seen from figure 4, adding seasonality of bioturbators (biomixers) improves simulation results, while adding seasonality of filter feeders into the simulation decreases the model performance. This is why All2 (now: all_bio_TS_1) was presented.
Indeed, the abiotic runs have no seasonality of benthos because they do not include benthos. We think it is valid to keep All2 since seasonality is exclusively linked to benthos.

The order of panels a-d will be changed according to the suggestion.

Figure 6:
The figure will be updated by using the new names of the experiment according to the suggestion.

Figure 7:
Panels will be reordered as suggested. All2 (now: all_bio_TS_1) is chosen because it has the least

RMSE. In figure 7 and 5, the aim is to compare the measurements to the best simulation (least RMSE) that could be achieved. This does not conflict with having no seasonality of benthos implemented in Ref1b and Ref1, because these runs are abiotic (see response to Figure 5). On the other hand, since we are comparing to measurements here, that inherit benthos seasonality, it makes sense to compare with a simulation which includes benthos seasonality.

The roman numbers indicating the five areas with pronounced changes are chosen to compare the measurements with the simulations. An indication of that will be added to the caption of the figure.

Figure 8:
We agree that it makes no sense to compare runs with different forcing. Thus the compared runs will be adjusted.

Indicating the amount of erosion and deposition in the plot is a very good idea. It will be added to the figure.

Figure 9:
The reviewer's suggestions will be adopted to update the figure.

Figure 10:
We are happy about the reviewers appreciation of this figure! The figure will be further improved according to the reviewer's suggestion.

**Line-by-line edits**

line 110: We will extend the description accordingly.

Line 120: The reference for figure 2b-f will be added.

Line 124: Common names will be added:

Line 161: How a and b are calculated is already described in the lines below and stated in the equations (5) and (6). However, we found a typo in equation (5). It is corrected as:
$$a_{bio} = 41.67 \cdot (1 + M_{TOT})^{0.34} \cdot (1 + B_{Indv})^{-0.09}$$

**Move paragraph starting at line 288 above previous paragraph:** Order of paragraphs will be switched

Line 321: A brief explanation on how to calculate the cosine similarity will be given.

Line 464-466: A further explanation will be added to the main text.

Line 85: Correction will be applied.

Line 93: Correction will be applied.

Line 157: Correction will be applied.

Line 365, 369, etc.: Correction will be applied.

Fig 7b: Indeed we meant "plot" and not measurements. This will be clarified in the caption of figure 7.

Line 541: Correction will be applied.

Line 545 and 551: Correction will be applied.

Line 552-553: Correction will be applied.

**Refernces used in the response letter:**

Adolph, W.: Praxistest Monitoring Küste 2008: Seegraskartierung: Gesamtbestandserfassung der eulitoralen Seegrasbestände im Niedersächsischen Wattenmeer und Bewertung nach EU-Wasserrahmenrichtlinie. NLWKN Küstengewässer und Ästuare, (2):1–62, 2010.

Becker, Marius. Suspended Sediment Transport and Fluid Mud Dynamics in Tidal Estuaries. PhD Thesis. University of Bremen, 2011.

Benninghoff, M., Winter C., 2019. Recent morphologic evolution of the German Wadden Sea. Sci. Rep. 9, 9293. https://doi.org/10.1038/s41598-019-45683-1.

Brückner, M., Schwarz, C., Coco, G., Baar, A., Boechat Albernaz, M. and Kleinhans, M.: Benthic species as mud patrol - modelled effects of bioturbators and biofilms on large-scale estuarine mud and morphology. Earth Surf. Process. Landforms. 46: 1128– 1144. https://doi.org/10.1002/esp.5080, 2021.

Daggers, T. D., Herman, P. M., and van der Wal, D.: Seasonal and Spatial Variability in Patchiness of Microphytobenthos on Intertidal Flats From Sentinel-2 Satellite Imagery. Frontiers in Marine Science, 7, 392. doi:10.3389/fmars.2020.00392, 2020.

Götschenberg, Axel; Kahlfeld, Andreas. The Jade. In: Die Küste 74. Heide, Holstein: Boyens. S. 263-274. 2008.

Kahlfeld, Andreas & Schüttrumpf, Holger. (2006). UnTRIM modelling for investigating environmental impacts caused by a new container terminal within the Jade-Weser Estuary, German Bight.

van Maanen, B., Coco, G., and Bryan, K. On the ecogeomorphological feedbacks that control tidal channel network evolution in a sandy mangrove setting. Proc. R. Soc. A. 471, 20150115. doi: 10.1098/rspa.2015.0115, 2015.

Ritzmann, A. and Baumberg, V.: Forschungsbericht 02/2013 – Oberflächensedimente des Jadebusens 2009: Kartierung anhand von Luftbildern und Bodenproben; NLWKN Niedersachsen, 2013.

Widdows, J. and Brinsley, M.: Impact of biotic and abiotic processes on sediment dynamics and the consequence to the structure and functioning of the intertidal zone. Journal of Sea Research, 48, 143-156. doi:10.1016/S1385-1101(02)00148-X, 2002.

---

## Referee Report (RR1)

Dear authors,
I want to acknowledge that the quantitative investigation of how benthos drives
morphological changes in tidal-dominated systems is a valuable contribution to the
community. I especially highlight the approach of multiple functional groups
individually and collectively influencing morphological changes.

Minor comments:

- L125-130: What species was attributed to the stabilizer functional group? It might
be obvious, but there is no mention of the species attributed to the seagrass
functional group either.

- The sinusoidal behavior of the seasonality shown in Table 3 is unclear. Is the
lowest seasonal value no biomass (complete die-off)? or certain minimal critical
biomass remains?

- There is no discussion on how seagrass can alter the composition of the sediment by
organic sediment accumulation beyond the hydrodynamic behavior.

- L550: I suggest the authors include a brief description of the conceptual model
introduced in Figure 10. I recommend making the arrow nomenclature consistent. For
instance, deposition/erosion is not clearly identified in the sketch. The subfigures
also lack alphabetic labeling.

---

## Author Response (AR2)

**Reply on RC3**

We thank Referee #3 for the constructive comments that have helped us to clarify and improve our manuscript. Our responses to the specific questions/requests (in **bold & *italic***) are listed below.

**Assessment:**
***L125-130:  What species was attributed to the stabilizer functional group? It might be obvious, but there is no mention of the species attributed to the seagrass functional group either.***

Thanks for pointing out that unclear message. We think the confusion is caused by naming this section *mapping of benthos* and then multiple species are mentioned with their functional group, except for Seagrass and Stabilizers. However, this section only deals with macrobenthos. The section title has been updated to clarify this. The species related to Seagrass and Stabilizers are mentioned multiple times in the main text (Seagrass: Lines 111, 220, 306; Stabilizers: Lines 198, 301). In order to increase clarity we have added two sentences summarizing all functional groups with their respective species in line 122-125:

*Biomixers and accumulators consist of macrobenthos while stabilizers are represented by a biofilm which is mainly assembled by microphytobenthos (MPB) of all contributing species. The seagrass present in Jade Bay belongs to the species Zostera noltii (Adolph, 2010).*

***The sinusoidal behavior of the seasonality shown in Table 3 is unclear. Is the lowest seasonal value no biomass (complete die-off)? or certain minimal critical biomass remains?***

In the original text, the sinusoidal behavior of the seasonality is first introduced in line 151-153 which simply refers to Table 3. Further details of the implementation are described in chapter 3.2.1 in line 193-196. To make the description clear in its first appearance, we have added a reference to section 3.2.1 in line 155-156.

***There is no discussion on how seagrass can alter the composition of the sediment by organic sediment accumulation beyond the hydrodynamic behavior.***

We have added discussion of this effect (see below) with two corresponding references in line 571-576:

*The impact of seagrass is prominent in close vicinity to the meadows but not so much within the meadow itself. One explanation is that the effect of organic sediment accumulation due to primary and detritus production and root and rhizome formation, which are main sources for sediment production (Garcia et al., 2003),  was not considered in this study. The found changes close to the meadows are in line with measurements indicating differences in bed level elevation between vegetated and non vegetated areas in the range of 3 cm per year (Potouroglou et al., 2017).*

***L550: I suggest the authors include a brief description of the conceptual model introduced in Figure 10. I recommend making the arrow nomenclature consistent. For instance, deposition/erosion is not clearly identified in the sketch. The subfigures also lack alphabetic labeling.***

We agree that deposition and erosion are not clearly distinguished in Figure 10. For this reason we have added an explanation for the red arrows in Fig. 10 (curved arrows describe sediment

redistribution (e.g. from tidal flats to tidal channels) while straight arrows indicate vertical erosion/deposition) and in the text Line 552-554 (see below):

*The different impacts of the mentioned functional groups in the Jade Bay are depicted in simplified form in Figure 10 where sediment redistribution (e.g. from tidal flats to channels) and vertical erosion/deposition patterns are distinguished.*

---

## Author Response (AR3)

**Reply on editors comment (Claire Masteller)**

We thank the editor for the constructive comments that have helped us to clarify and improve our manuscript. Our responses to the specific questions/requests (in **bold & *italic***) are listed below.

**Assessment:**

***I agree with the reviewers that while the model names and table provide the reader with some insight into the differences between model runs, it can be difficult to keep track of. I appreciate that the authors added more details to Table 3 to clarify the naming conventions of the model. This has been implemented for some, but not all of the figures. I do encourage the authors consistently implement the reviewers suggestion to add more descriptive model titles in the figures of the paper, specifically the maps of model output.***

The purpose of changing the names was to give the reader a better overview of what is depicted in the respective figure. In figure 8 and 9 we think the short text is better suited in the image than writing *run_abio_no_TS_10 - acc_no_TS_10* and instead have added *run_abio_no_TS_10 - acc_no_TS_10* into the caption of the figure.

***Table 1 clarifies that the parametrization comes from observational data, but it is unclear to me from my reading what observational data is used to inform model forcing. Additional details on this are necessary to indicate how storms may modify different aspects of the model.***

Water elevation data was used which is indicated in supplementary material. We have added this information to Table 1.

***Further, given that model values of pd, gd, ps, and gs are fundamental to the implementation of the model, providing the range of values used across model runs for these parameters would strength the contribution and methods section. The authors currently point to existing citations from other works, but providing further details on the range of expected values related to the parameterization and implementation of the model in this contribution would be helpful to a reader trying to reproduce this study.***

We have extended the explanation and added a new formula to make the calculation comprehensible for the reader:

*The other biomixing function $p_d$ is calculated following Brückner et al. (2021), which is also based on the data from Cozzoli et al. (2019). Abiotic ($\tau_c^0$) and biotic critical shear stress for erosion ($\tau_c^{bio}$) are defined based on the respective $\tau_b$ value at which a minimal erosion rate of 25 g m-2 is reached. This is done by converting formula (3) into:*

$$\tau_c = b - c \cdot log\left(\frac{a - R_{25}}{R_{25}}\right) \qquad (8)$$

*$\tau_c^0$ is calculated using $a_0$, $b_0$, and $c_0$ which are constants for the defaunated control experiments given in Table 1 in Cozzoli et al. (2019). For $\tau_c^{bio}$ $a_{bio}$, $b_{bio}$, and $c_0$ are used. $p_d$ is then calculated via:*

$$p_d = \frac{\tau_c^{bio}}{\tau_c^0} \qquad (9)$$

$$(9)$$

***Throughout - please provide units for all variables at the first instance that they are introduced.***

Units for all variables at the first instance that they are introduced are now provided in the revised version.

**L160 - For equations 1 and 2. Are A and B fractional values between 0 and 1 or dimensional quantitates? For these stabilizing functions, are there more complex functions underpinning these or do A and B act purely as scalars? I see that these relationships are defined in the following sections, but a clarifying sentence/short paragraph at the end of this section and priming the reading for the progression to the next section would improve readability.**

A is the abundance/number of individual and B is the biomass in mg AFDW. Both are calculated with the species abundance model. The way B is used in order to calculate the stabilizing functions is described in equation (4), (5) and (6). We forgot to explain how A is used, and an explanation is now added in the text. The description of the quantities A and B is also added in the text according to the suggestion.

**L165-L170 - Is Rtot different that Er in Equation 2? I am unsure if these are the same variable or if there is some component of erosion specifically attributed to biomixing effects. Please add a note to clarify or check for consistency.**

The meaning and difference between $R_{tot}$ and $E_r$ become clear when looking at the dimensions (added in the revised version). $E_r$ is a rate dependent on time and $R_{tot}$ refers to the result of measurements by Cozzoli et al. (2019), which describes the total amount of sediment eroded within a defined time span. $R_{tot}$ is used to derive the biomixing function $g_d$, which is then fed into Equation 2 to calculate $E_r$. We have added a short explanation in the revised text:

*In our model, the formulae from Cozzoli et al. (2019) are adopted to relate biomixing effect with the overall metabolic rate $M_{TOT}$ (mW). In this study measurements of the total eroded sediment per unit area in a given time, $R_{TOT}$ ($\frac{g}{m^2}$), were taken. Assuming that the erosion rate ($\frac{kg}{m^2 s}$) over the given time is constant it can be described by: $R_{TOT} = \frac{a}{1 + exp\left(\frac{b - \tau_b}{c}\right)}$*

**L198 - Are the range of raw values used for these terms useful to report or provide here for reproducibility?**

We have made this point more clear in order to ensure reproducibility:

*To account for this seasonal variability, a multiplication factor for $M_{TOT}$ was introduced according to a sine function with a period of 1 year, reaching the maximum value of 1.0 in summer and the minimum of 0.1 during winter.*

**204 - When erosion rate is used here is this related to a variable in the model? I see that gs is set to 1, and that gs modifies Er, but is this directly reflecting the statement regarding erosion rate, or is this a consequence of the assumption that in the winter months, both ps and gs are set to 1 and Er = Er0? Please clarify how these are related in the main text.**

Yes, the modification of $g_s$ and $p_s$ directly reflect the statement regarding erosion rate and critical shear stress. We have modified the respective paragraph to make this clear.

**L213 - In Equation 9 a new biomass variable has been introduced. Is this variable separate from B, or the sum of, Bindv, or consistent with these treatments? Please clarify.**

$B$ is biomass of macrobenthos in general. Since the effects of bio-accumulators is scaled with the biomass of bio-accumulators, a new symbol $S$ was used. To avoid confusion we have now changed it to $B_{acc}$ for consistency in use of symbol for biomass. Explanation is provided.

***L206-L225 - Are the affections of accumulators and seagrass explicitly accounted for in the model framework presented in equations 1 and 2? I recognize that there are model parameterizations associated with these terms, but am unclear on whether or not these terms are accounted for in ps, gs, pd, and gs, or as separate parameterizations. Table 1 is somewhat helpful in clarifying these, but similar clarifications should be made in the text for readability. For example, a sentence clarifying that seagrass hydrodynamics are accounted for in SCHISM using an existing module would help the read follow the model implementation more clearly. While these are included in the table, also reiterating this in the sections where each element of the model are introduced would strength the methods.***

The effect of accumulators is in sediment settling velocity and the effect of seagrass is in turbulence and bottom drag. They are not explicit accounted for in equation (1) and (2). We have modified the first paragraph of 3.2 to clarify this:

*"Impacts of benthos on sediment are formulated through scaling functions between benthos abundance/biomass and model parameters for sediment dynamics, namely the critical shear stress for erosion $\tau_c$ (Pa), the erosion rate $E_r$ ($\frac{kg}{m^2 s}$), the sediment settling velocity $W_{sed}$ ($\frac{mm}{s}$) and hydrodynamic parameters for turbulence and bottom shear stress. For sediment erosion, the general approaches by Knaapen et al. (2003) for $\tau_c$ and Paarlberg et al. (2005) for $\tau_c$ and $E_r$ are applied. An abiotic critical shear stress for erosion $\tau_c^0$ and erosion rate $E_r^0$ are scaled by dimensionless biomixing functions $p_d$, $g_d$ and stabilization functions $p_s$, $g_s$, respectively, which depend on abundance $A$ (number of individuals) and biomass $B$ ($mg$ ash free dry weight (AFDW)) of these two functional groups:*

$$\tau_c = \tau_c^0 \cdot p_d(B, A) \cdot p_s(B, A) \tag{1}$$

$$E_r = E_r^0 \cdot g_d(B, A) \cdot g_s(B, A) \tag{2}$$

*Changes in hydrodynamics by the effect of seagrass are incorporated using the submerged aquatic vegetation model (SAV) of SCHISM (Zhang et al., 2016) and changes in $W_{sed}$ by the effect of accumulators are applied according to a filter feeder ingestion rate model (US Army Corps of Engineers, 2000). Both are explained in following sections. "*

***L302 - "lies below 20% deviations from the measurements for the majority of the stations" os a bit confusing. Rephrase for clarity.***

We have rephrased this sentence and added two sentences:

*"To assess the performance of the decision tree-based SAM model, the measured data were split into training and validation datasets. The training dataset was used for training the model and the validation dataset was checked against the resulting estimations of biomass and abundance. The performance of the SAM varies among the selected species. For the majority of the data points, the estimated value deviates from the measured value by less than 20% (Fig. S2, supplementary). "*

***L307 - missing a space between "the years 2008...."***
***L411 - Typo, should be "Deposited"***
***L452 - Typo, should be "biomixers"***

*L486 - modify to "in the direct vicinity"*
*Figure 7 has a typo in "measurements"*

Corrected according to comment.

*Fig. 5, 8, - I totally missed the bar plots in the bottom right on first reading, I would recommend making these larger and labeling the axes to draw the readers attention and better connect the maps to the main channel plot. I think these are quite useful to provide context and aids in the interpretability of the model output maps.*

We have increased the size of these plots and labelled the axes according to the editors' suggestion.

*Fig. 7., 9 This figure has quite a bit of information in it. In terms of clarity and comparison, the author may consider adding a new figure or fifth panel that more directly compares the net changes in each of the identified regions to one another across the model types. I recognize that this is discussed in the main text of the paper, but an additional summary visual may reinforce the point that different model runs are capturing morphologic change in different regions of the study site with varying degrees of performance/variability. Perhaps a box plot of the changes of each model cell in each region, so all three model runs can be compared directly to observations in Region 1, and so on and so forth for each region?*

We have added a new panel e) to both figure 7 and 9. In Figure 9 we have followed the editor's recommendation by adding a boxplot diagram. In Figure 7, we found that a boxplot diagram with a small number of possible values [-2, -1, 0, 1, 2] looked confusing since it shrinks to just a bar in some cases. Therefore we have adopted a violin plot which provides a probability distribution to better reflect the variability. References to figure 7e and 9e have been added to the main text.